



# A gridded multi-site precipitation generator for complex terrain: An evaluation in the Austrian Alps

Hetal Dabhi[1], Mathias W. Rotach[1], and Michael Oberguggenberger[2]

[1]Department of Atmospheric and Cryospheric Sciences, University of Innsbruck, Innsbruck, Austria
[2]Unit for Engineering Mathematics, University of Innsbruck, Innsbruck, Austria

**Correspondence:** Hetal Dabhi (Hetal.Dabhi@student.uibk.ac.at)

**Abstract.**

For climate change impact assessment, many applications require very high-resolution data of precipitation consistent both in space and time for current as well as future climate. In this regard, stochastic weather generators are designed as a statistical downscaling tool that can provide such data. Here, we adopt the framework of a precipitation generator of Kleiber et al.

(2012), which is based on latent and transformed Gaussian processes, and propose an extension to that for a mountainous region with complex topography. The model is used to generate two-dimensional fields of precipitation with 1 km spatial and daily temporal resolution in a small region with highly complex terrain in the Austrian Alps. This study aims at evaluating the model for its ability to simulate realistic precipitation fields over the region using historical observations from a network of 29 meteorological stations as an input, discusses its added value over the original set-up and its limitations. Results show that

the model generates realistic fields of precipitation with good spatial and temporal variability. The model is able to generate some of the difficult areal statistics useful for impact assessment such as areal dry and wet spells of different lengths and areal monthly mean of precipitation with great accuracy. The model also captures the inter-seasonal and intra-seasonal variability very well while the inter-annual variability is well captured in summer but largely underestimated in autumn and winter. The proposed model adds substantial value over the original modeling framework, specifically for the precipitation amount. The

model is not able to reproduce realistic spatio-temporal characteristics of precipitation in autumn. We conclude that with further development, the model is a promising tool for downscaling precipitation in complex terrain for a wide range of applications in impact assessment studies.

## 1   Introduction

Precipitation is a major component of the hydrological cycle. With global warming, the hydrological cycle is expected to intensify and the risk associated with extreme events will increase (Tabari (2020); Pfahl et al. (2017)). The resulting changes in the precipitation will be unequally distributed around the world. There are many hydrologic responses to climate change and the potential impacts of these are likely to affect availability of fresh water, agriculture, timing and severity of wildfires and habitat sustainability (Bates et al. (2008); Kundzewicz et al. (2008)). With the increasing awareness about climate change and

its global impact on the ecosystem and human societies (Konapala et al. (2020); Haddeland et al. (2014); Schewe et al. (2014)),





there is also an increasing need to understand the effects and impacts that would occur at local scale. Knowledge on how the local hydrological cycle and water resources will be affected by climate change is essential for planning reliable adaptation strategies and water policy.

In Austria, where a large part of the country is covered by mountains, the local hydrological cycle depends widely on temporal and spatial variations in precipitation. Tourism and agriculture are among the main drivers of Austria's economy. Accessibility of water resources for human consumption and ecosystems largely depends on the spatio-temporal distribution of precipitation. In the Austrian Alps, studies on the observed as well as projected impact of climate change show the changes on the availability of snow cover as well as water flux (e.g. Abermann et al. (2009); Wijngaard et al. (2016)). This ultimately will have impact on

the economy, the ecosystem, the environment and the society. To assess the impacts of climate change at local scales, precise climate information is critical which can serve the needs of the decision makers. Often such information should be consistent in space and time for present as well as future climate. Many applications in hydrology require a very high resolution data of precipitation, typically of 1 km for spatial and daily for temporal scale. However, obtaining such high-resolution precipitation data is still a challenging task especially in the mountainous region (Henn et al. (2018)). Most importantly for the complex

topography such as the Austrian Alps, even 1 km resolution cannot include the impact of topography on climate correctly. For such regions, many applications in hydrology and ecology require even higher resolution data – in the spatial scale of 100 m and hourly for temporal scale. Climate models with higher resolutions like Regional Climate Models are also unable to provide such data. For that reason, various downscaling methods have been in use in the past few decades. Among all the downscaling methods, statistical downscaling using stochastic weather generators (WGs) has become very popular mainly because WGs

are computationally parsimonious.

A vast variety of WGs have been developed based on different approaches. Most widely used WGs are based on a rather simplistic approach in which the sites are mutually independent in space and time. Such WGs are generally referred to as single-site WGs. Among the single-site WGs, the most popular WGs are the parametric models based on Richardson (1981),

who used a Markov chain to simulate time series of precipitation occurrence (wet/dry days) and amount and other variables were generated upon the condition of whether the generated day is wet or dry (e.g. Dabhi et al. (2021), Caron et al. (2008); Zhang et al. (2004); Dubrovský et al. (2004); Wilks (1992)). The details on the available WGs can be found in the review articles by Ailliot et al. (2015); Maraun et al. (2010); Wilks and Wilby (1999) where Maraun et al. (2010) focused solely on precipitation.


The major drawback of single-site WGs is that they are focused on a single location only, which can generate realistic data at a location but lacks the spatially correlated structure in the generated data. Getting a spatially and temporally consistent dataset – which is more realistic – from single-site models is impossible. For that reason, in the past two decades the focus has moved towards development of spatio-temporal WGs also known as multi-site WGs. As for precipitation, given the uneven nature of

its occurrence and intensity, it becomes more challenging to model it by keeping the spatio-temporal structure. In particular, in



complex topography like in the Alps, this task is even more challenging.

Numerous approaches have been proposed to generate spatially and temporally correlated precipitation data. Wilks (1998) did one of the early works on multi-site generation of daily precipitation where single-site parametric WGs at sites were forced with correlated random numbers to generate the occurrence of precipitation and the amount of precipitation was generated using a mixture of two exponential distributions. Other approaches to spatio-temporal modeling of precipitation are hidden Markov models (e.g. Ghamghami et al. (2016); Charles et al. (1999); Ailliot et al. (2009)), copula-based approaches (Serinaldi (2009); Bárdossy and Pegram (2009)), resampling based on k-nearest neighbours (Apipattanavis et al. (2007); Buishand and Brandsma (2001); Rajagopalan and Lall (1999)), Poisson cluster models (Ramesh et al. (2012); Cowpertwait (1995)), artificial neural networks (Harpham and Wilby (2005)) and Generalised Linear Model (GLM) based approaches (Kleiber et al. (2012); Verdin et al. (2018)). Baxevani and Lennartsson (2015) used spatio-temporal model using censored latent Gaussian field for precipitation generation. Olson and Kleiber (2017) used the Approximate Bayesian Computation method. Gao et al. (2021) developed a multi-site stochastic daily rainfall model by coupling a univariate Markov chain with a multi-site rainfall event model. There also exist more sophisticated WGs that can provide high-resolution spatio-temporal fields combining physical and stochastic approaches (e.g. Peleg et al. (2017); Paschalis et al. (2013)).

However, most of the aforementioned approaches simulate precipitation only at the locations where observations are available and such multi-site WGs have been implemented for the Alps. For example, Keller et al. (2015) and Keller et al. (2017) used a Wilks-type WG for precipitation simulation and downscaling, respectively, for a mountainous catchment in the Swiss Alps. Breinl et al. (2013) used a semi-parametric multi-site precipitation generator for the mountains in the Austrian-German Alps. However, high resolution data in space and time are needed to provide more realistic input for local climate impact assessment. To achieve that, a gridded multi-site model is required. Sparks et al. (2017) proposed a multi-site multivariate WG based on the use of periodically extended empirical orthogonal functions (EOFs), in which they modeled precipitation as a censored latent Gaussian process. They generated gridded data of precipitation and temperature over Europe using gridded input data, but their WG can not provide gridded data without gridded observations. Peleg et al. (2017) developed a WG called AWE-GEN-2d and used it in the Swiss Alps, which can generate two-dimensional fields of various meteorological variables where precipitation is generated at 2 km spatial and 5 min temporal resolution. Although their approach is sophisticated as it is a hybrid approach combining dynamical and statistical approaches, it requires spatially distributed data for calibration of the WG and it cannot generate data on a region outside the calibration region. Such WGs are of limited use if the observed gridded data are not available which is often the case. To our knowledge, not much work has been done for complex terrain like the European Alps using multi-site gridded WG without gridded observations.

Wilks (2009) developed one of the first ever WGs which can provide gridded data of precipitation and also temperature at locations with no observations. Kleiber et al. (2012) also gave an approach using the GLM based model which uses Gaussian processes to generate gridded data. Their approach is appealing as it generates the readily available field of precipitation





using kriging for interpolation of the model parameters. The advantage of their approach is that in the GLM framework, one could include various covariates such as large-scale climate indices, local climate information, topographical information etc. which makes the model more flexible. Another advantage is that it is a probabilistic approach which allows one to quantify the uncertainties in the parameter estimation. However, Kleiber et al. (2012) tested the model only for the multi-site precipita-

tion generation, i.e. at locations with observation and not for the generated gridded data of precipitation. Verdin et al. (2018) modified the framework of Kleiber et al. (2012) by including seasonal precipitation as an additional covariate and evaluated the model for gridded data but it was implemented in the flat terrain. Also, their modified model and the original model, both used an isotropic and stationary covariance structure with ordinary kriging (OK) for the interpolation of the model parameters which may not be suitable for the complex topographical terrain. Bennett et al. (2018) generated precipitation fields using a

latent-variable approach that provides a parsimonious method to jointly generate rainfall occurrence and amount. They used an isotropic powered exponential function for including spatial correlations and kriging for the interpolation of the parameters. However, they also implemented their model in relatively flat terrain in South Australia. To our knowledge, no space-time gridded precipitation generator has been evaluated for its ability to generate two-dimensional fields of precipitation in the highly complex terrain without needing gridded input data.


Here, we propose an extension to the framework of Kleiber et al. (2012) for complex terrain and evaluate the model for its capabilities to generate realistic two-dimensional fields of precipitation for a mountainous region in the Austrian Alps. In addition, we examine the added value of our model to the original isotropic set-up and discuss the limitations of the model.

This article is organized in a following way: Section 2 describes the extension to the isotropic framework for the implementation in a mountainous region. Section 3 describes the study area and the data used in the study and model evaluation strategy. Section 4 discusses the results, Section 5 entails the discussion on the results and Section 6 summarizes the study.

## 2  Model Description

### 2.1  Precipitation Occurrence

At a location $s$ and on a day $t$, precipitation occurrence $O(s,t)$ is 0 (dry day) if no precipitation and 1 (wet day) if precipitation occurs. A wet day is defined when the precipitation amount exceeds 0.1 mm.

For a set of locations $\mathbf{s}$ and on a day $t$, a latent Gaussian process $W_O(\mathbf{s},t)$ is defined with mean function $\mu_O(\mathbf{s},t)$ and covariance function $C_O(\mathbf{h},\mathbf{v},t)$, where $h = |s_i - s_j|$ is the horizontal (Euclidean) distance between two locations denoted by $i$ and $j$

and $v = |v_i - v_j|$ is the elevation difference between the two locations. The suffix 'O' stands for occurrence. Since we are implementing the model in a mountainous region, we define the covariances among the sites as a function of the difference in the elevation, too. In comparison with the original model where an isotropic covariance structure is used as a function of horizontal distances among the sites, this allows us to include anisotropy in the model. The precipitation occurrence then is





defined as

$$O(\mathbf{s},t) = 0 \quad if \quad W_O(\mathbf{s},t) < 0$$
$$O(\mathbf{s},t) = 1 \quad if \quad W_O(\mathbf{s},t) \geq 0 \tag{1}$$

where the mean function is

$$\mu_O(\mathbf{s},t) = \beta_O(\mathbf{s})' X_O(\mathbf{s},t) \tag{2}$$

'$X_O$' is a vector of covariates and '$\beta_O$' is a vector of regression parameters as in Eq. 5.

Kleiber et al. (2012) used a stationary and isotropic exponential covariance function of the form

$$C(\mathbf{h},t) = exp\left(\frac{-|\mathbf{h}|}{A(t)}\right) \tag{3}$$

where $A(t)$ is the time dependent scale parameter.

Since our goal is to use the model in complex topography, we introduce anisotropy in the covariance function (Eq. 3) by taking the difference in elevation between two locations. Thus, our stationary and anisotropic covariance function $C(\mathbf{h},\mathbf{v},t)$ takes the following form

$$C_O(\mathbf{h},\mathbf{v},t) = exp\left(-\frac{|\mathbf{h}|}{A(t)} - \frac{|\mathbf{v}|}{B(t)}\right) \tag{4}$$

where $A(t)$ and $B(t)$ are the time dependent scale parameters in the horizontal and vertical direction, respectively.

At the base of this model is the single-site precipitation generator based on the GLM framework (e.g. Stern and Coe (1984); Chandler and Wheater (2002); Furrer and Katz (2007)) which is similar to a Richardson-type precipitation generator (Richardson (1981)), where daily precipitation occurrence is modeled as a first-order Markov chain and daily precipitation amount is modeled using a gamma distribution. The GLM-based approach allows more flexibility as one may include as many covariates as desirable, through which the seasonality or the influence of large scale circulation on the local precipitation can be included. In the GLM approach, taking the previous day's occurrence as a covariate, forms a first-order Markov chain. Thus, at individual sites, the model reduces to a logit model which is given by

$$log\left(\frac{p_t}{1-p_t}\right) = \beta_O' X_O \tag{5}$$





where, $p_t$ is the probability of occurrence on a day $t$.

Note that we use a logit link function instead of a probit link function, as was the choice in the original model.
$C_O(\mathbf{h}, \mathbf{v}, t)$ is estimated as the covariance matrix of the residuals in the logit model. We use a method of moments approach as suggested by Kleiber et al. (2012) to estimate the scale parameters $A$ and $B$. The parameters are estimated separately for each month to allow seasonality in the generated data.

The Gaussian process itself provides a spatial interpolation method 'kriging' so that the model parameters $\beta_O$ associated with each covariate, which are estimated at observation locations, can be interpolated to any location of interest. These gridded coefficients are then used to obtain the mean function (Eq. 2). Here, we use kriging with external drift (KED) (see e.g. Wackernagel (2003)) to interpolate the regression parameters. Since precipitation in the mountains is unequally distributed across the terrain, we allow elevation as an external drift in kriging such that the predicted values of precipitation (through the interpolated parameters) reflect the elevation dependency of precipitation. In KED, an auxiliary variable is assumed, which is elevation here, that is linearly related to the variable of interest which is the $\beta$ parameter associated with each covariate in the model.

We also compare the results of our model with a simulation using ordinary kriging (OK) instead of KED in our model and also with the original isotropic model using OK and KED which will be discussed in the Section 4.3.

## 2.2 Precipitation Amount

To simulate spatially correlated fields of precipitation, another Gaussian process $W_A(\mathbf{s}, t)$ is defined with mean function $\mu_A(\mathbf{s}, t)$ and covariance function $C_A(\mathbf{h}, \mathbf{v}, t)$, such that

$$Y(\mathbf{s}, t) = G_{\mathbf{s}, t}^{-1}(\Phi(W_A(\mathbf{s}, t)) \tag{6}$$

where $G_{\mathbf{s}, t}$ is the cumulative distribution function (CDF) of the gamma distribution at the locations $\mathbf{s}$ and time $t$, and $\Phi$ is the CDF of a standard normal distribution. At an individual location, the amount model is the gamma GLM with logarithmic link function as given by Furrer and Katz (2007). The shape parameter of the gamma distribution varies with space but not with time, while the scale parameter varies with both space and time. This way each location has its own distinct value of shape and scale parameters, with the scale parameter varying with time. The scale and shape parameters are estimated at each individual observation site using the maximum likelihood approach and then are interpolated using KED. We allow the scale parameter to vary with every month to include seasonal variations in precipitation at each location. The mean function $\mu_A(\mathbf{s}, t)$ is again obtained from a regression on covariates. The covariance function $C_A(\mathbf{h}, \mathbf{v}, t)$ is the same as given in the occurrence model (Eq. 4), but with different parameters. The parameters of the covariance function are estimated for each month separately to allow the seasonality in spatio-temporal pattern of precipitation amount.



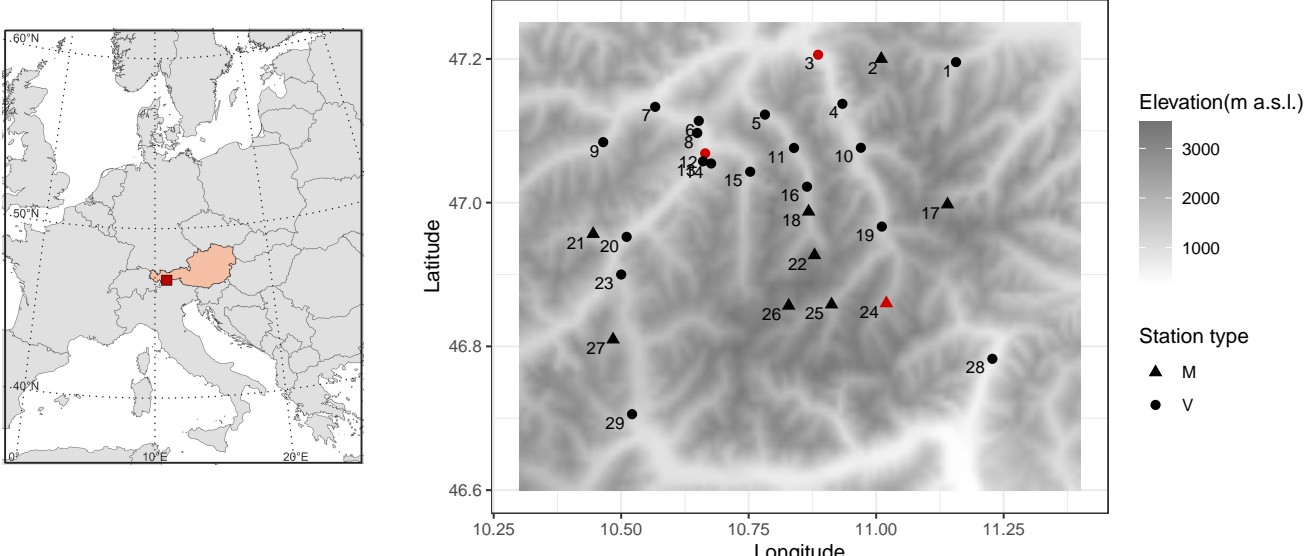

**Figure 1.** Study area showing the locations of the 29 meteorological observation stations (right) whose data are used in the study and the location of the region in the central Alps (left). Longitude and latitude are given in degrees east and north, respectively. Gray shading denotes the elevation (m a.s.l.). Stations with elevation higher than 1500 m a.s.l., usually high-mountain stations, are shown by the symbol type M and the stations with elevation less than 1500 m a.s.l., typically the valley stations, are shown by symbol type V. The stations shown in red colour are selected as example stations to illustrate the results in the article.

## 3 Implementation

### 3.1 Study area and data

The model is implemented in a small region comprising highly complex terrain (ranging from 256 m a.s.l. to over 3500 m a.s.l.) in the Austrian Alps. The area is surrounding the catchment of River Oetz, mainly in the federal state of Tyrol but also includes a part of the Autonomous Province of South Tyrol in northern Italy. The reason for selecting this region is that the catchment of river Oetz is a widely researched area (e.g. Wijngaard et al. (2016); Abermann et al. (2009)). To include more stations in the study, we allow stations from the surrounding region, also including northern Italy. The study region is comprised of several valleys including Oetz and Pitz Valley in Austria and Passeier Valley in South Tyrol. Observation data from 29 meteorological stations (Figure 1) based on the availability of homogeneous time-series of 30 years period from 1981 to 2010 are selected. The dataset comprises data provided by the Austrian National Weather Service (ZAMG – Zentralanstalt für Meteorologie und Geodynamik), the Hydrographic Service of Austria, the Hydrographic Service of the Autonomous Province of South Tyrol, the Institute of Atmospheric and Cryospheric Sciences – University of Innsbruck and TIWAG (Tiroler Wasserkraft AG). The highest station is on a glacier (Hintereisferner) at the elevation of 2860 m a.s.l., while the lowest is at 588 m a.s.l in northern Italy. A few stations have missing values for single days or a short period. Our model ignores such values while computing the





observed statistics. The data at all the stations are quality controlled by the respective service providers.

In the northern part of the region, we have a dense network of stations, while the southern part has relatively fewer stations. The average inter-station distance between two locations is 28.15 km. The maximum inter-station distance is 72.84 km and the minimum inter-station distance is 1.25 km. The average altitude difference between two stations is 0.605 km, while the maximum altitude difference is 2.272 km. The locations of the 29 stations are shown in Figure 1 and further details about the stations are given in Table 1.


The mean annual precipitation observed in the lowlands is approximately 780 mm on an average of 150 wet days in a year, while the highest mean annual precipitation is observed at the high-mountain station Dresdner Huette which is 1320 mm with an average of 176 wet days in a year. The highest number of mean annual wet days is 220 at St. Martin in Passeier Valley in South Tyrol with 887 mm mean annual precipitation.


Due to strongly different topography, a large variability both in space and in time is observed in the dataset. Out of all the 29 stations, Prutz has the most distinct climatological characteristics. For example, Prutz has the largest variability in almost all the months. Also, the most extreme precipitation (156.5 mm) in a day is recorded at Prutz (in July 2009) whereas amongst the remaining 28 stations, the second highest amount of precipitation recorded on the same day was at Dresdner Huette (35.1

mm). Apart from Prutz, only Dresdner Huette recorded a daily precipitation amount as high as 120.4 mm in the 30 years of record. At the location St. Leonhard im Pitztal, there are two stations operated by two different service providers. One is by the hydrographic service of Austria (St. Leonhard im Pitztal-1, see Table 1) in the northern part of the Valley and the other by ZAMG (St. Leonhard im Pitztal-2, see Table 1) in the southern part of the Pitz Valley. St. Leonhard im Pitztal-2 has somewhat different climatological characteristics than the nearby stations. Another station is St. Martin which has quite different

climatologies compared to the Austrian stations. Note that this station is in the south of the Alpine crest (i.e. in northern Italy) and has the lowest elevation in the observed data. Thus, there are high variations in the observed climatologies of precipitation from valley to valley and also for stations within the same valley. This adds a particular challenge to the simulation.

To reduce uncertainty and add more robustness to the observations, we increase the sample size of the observed data by con-

sidering a 7-days window centred at the day of interest. We generate N=30 stochastic realisations, each 30 years long (30 realisations×30 years = 900 years), of daily two-dimensional fields of precipitation on a 1 km grid over the region using the aforementioned observed daily data of 30 years. The Shuttle Radar Topography Mission (SRTM) 1 km (30 arc second) resolution dataset (Becker et al. (2009)) is used as a simulation grid. We select 1 km spatial resolution to reduce the simulation time as well as data storage requirement.






**Table 1.** List of the 29 meteorological stations whose data are used in the study. The names in **bold** letters are the three representative stations to illustrate the results.

| Id no. | Name of the station | Longitude | Latitude | Altitude (m a.s.l.) |
|---:|---|---:|---:|---:|
| 1 | Gries im Sellrain | 11.16 | 47.20 | 1200 |
| 2 | Kuehtai | 11.01 | 47.20 | 1970 |
| 3 | **Oetz** | **10.89** | **47.21** | **760** |
| 4 | Umhausen | 10.93 | 47.14 | 1040 |
| 5 | Jerzens-Ritzenried | 10.78 | 47.12 | 1120 |
| 6 | Fliess | 10.65 | 47.11 | 860 |
| 7 | Landeck | 10.57 | 47.13 | 800 |
| 8 | Ladis-Neuegg | 10.65 | 47.10 | 1350 |
| 9 | See im Paznaun | 10.46 | 47.08 | 1040 |
| 10 | Laengenfeld | 10.97 | 47.08 | 1180 |
| 11 | St. Leonhard im Pitztal-1 | 10.84 | 47.08 | 1329 |
| 12 | **Prutz** | **10.66** | **47.07** | **871** |
| 13 | Ried im Oberinntal | 10.66 | 47.06 | 895 |
| 14 | Fendels | 10.68 | 47.05 | 1343 |
| 15 | Kaunertal-Vergoetschen | 10.75 | 47.04 | 1269 |
| 16 | St. Leonhard im Pitztal-2 | 10.86 | 47.02 | 1460 |
| 17 | Dresdner Huette | 11.14 | 47.00 | 2290 |
| 18 | Plangeross | 10.87 | 46.99 | 1605 |
| 19 | Soelden Schmiedhof | 11.01 | 46.97 | 1380 |
| 20 | Pfunds | 10.51 | 46.95 | 992 |
| 21 | Spiss | 10.45 | 46.96 | 1540 |
| 22 | **Pitztaler Gletscher (Pitztal Glacier)** | **10.88** | **46.93** | **2860** |
| 23 | Nauders | 10.50 | 46.90 | 1360 |
| 24 | Obergurgl | 11.02 | 46.86 | 1940 |
| 25 | Vent | 10.91 | 46.86 | 1890 |
| 26 | Vernagtbach | 10.83 | 46.86 | 2640 |
| 27 | Ausserrojen | 10.48 | 46.81 | 1833 |
| 28 | St. Martin im Passeier Beobachter | 11.23 | 46.78 | 588 |
| 29 | Marienberg | 10.52 | 46.71 | 1310 |

For the Northern Atlantic Oscillation Index (NAOI) (see Section 3.2), a daily time series from 1981 to 2010 is obtained from the National Oceanic and Atmospheric Administration (NOAA) website[1].

---

[1] https://www.cpc.ncep.noaa.gov/products/precip/CWlink/pna/nao.shtml





It should be noted that the observed data are from different service providers, therefore the time of the data collection may
differ which may affect the results.

## 3.2  Selection of covariates in the model

We allow several covariates in the model such that the model can capture a realistic structure of precipitation patterns over the
region. This includes the day-to-day time dependence, seasonality as well as the influence of large scale circulation. As the
first covariate, we select previous day's occurrence ($\mathrm{Occ}_{t-1}$) as possible covariate such that day-to-day temporal dependency
in occurrence at a location is captured. To include seasonality, time dependent $1^{st}$ and $2^{nd}$ order harmonics of sine and cosine
(see Table 2) are considered as the possible covariates. To allow the influence of large scale circulation over Europe, the NAOI
is considered as a possible covariate. Studies show that there are links between the NAOI and precipitation characteristics
(Casty et al. (2005); Beniston (1997)). A strongly positive NAOI is associated with persistent high pressure over the alpine
region, resulting in warmer than average temperatures and lower than average precipitation. In general, winter NAOI correlates
negatively with precipitation. Along with the described covariates, we also take into account their interaction terms as possible
covariates.

For the selection of the covariates in the model, we use both the Akaike Information Criterion (AIC) (Akaike (1974)) and
the Bayesian Information Criterion (BIC) (Schwarz (1978)). None of the two criteria selects the same set of covariates at all
the stations. BIC has a tendency to select the simplest model whereas AIC has a tendency to select more complex models.
However, it turns out that BIC helps in identifying the most important covariates at all the stations.

The three most important covariates for the occurrence model at the majority of stations are $\mathrm{Occ}_{t-1}$, $\mathrm{Cos}(2\pi t/n)$ and $\mathrm{Cos}(4\pi t/n)$,
where '$t$' is the day of year. We select those covariates which are selected by both AIC and BIC at the majority of the stations.
The selected covariates are listed in Table 2. BIC selected this set of covariates at 18 stations out of total 29 stations (see Section
3.1), while AIC selected the same set of covariates at 11 stations. Thus, the vector of covariates in the model is

$$
\begin{aligned}
X_O(s,t) = (1, \mathrm{Occ}_{t-1}, \mathrm{Cos}\left(\frac{2\pi t}{n}\right), \mathrm{Sin}\left(\frac{2\pi t}{n}\right), \mathrm{Cos}\left(\frac{4\pi t}{n}\right), \mathrm{Cos}\left(\frac{4\pi t}{n}\right) * \mathrm{Sin}\left(\frac{4\pi t}{n}\right), \\
\mathrm{Occ}_{t-1} * \mathrm{Cos}\left(\frac{2\pi t}{n}\right), NAOI)
\end{aligned}
\tag{7}
$$

where '$n$' is 365 or 366 in case of a leap year. The first term is associated with the intercept in the model.


For the precipitation amount also, we take into account all the possible covariates as described for the occurrence model. We
select the covariates using both AIC and BIC for the amount model also. Additionally, selecting the same 7 covariates as in
the occurrence model at the majority of stations, the $2^{nd}$ harmonic of sine is also selected by both AIC and BIC (17 stations
by BIC and 16 stations by AIC). Thus, we allow a total of 8 covariates in the amount model (see Table 2) and the vector of





**Table 2.** List of the covariates included in occurrence and amount model (Eq. 7 and Eq. 8)

| No. | Name of the Covariate | Description | Occurrence model | Amount model |
|-----|----------------------|-------------|------------------|--------------|
| 1 | $Occ_{t-1}$ | Previous day's occurrence | ✓ | ✓ |
| 2 | $Cos(2\pi t/n)$ | 1st harmonic of Cosine | ✓ | ✓ |
| 3 | $Sin(2\pi t/n)$ | 1st harmonic of Sine | ✓ | ✓ |
| 4 | $Cos(4\pi t/n)$ | 2nd Harmonic of Cosine | ✓ | ✓ |
| 5 | $Sin(4\pi t/n)$ | 2nd harmonic of Sine | × | ✓ |
| 6 | $Cos(4\pi t/n) * Sin(4\pi t/n)$ | Interaction of $Cos(4\pi t/n)$ and $Sin(4\pi t/n)$ | ✓ | ✓ |
| 7 | $Occ_{t-1} * Cos(2\pi t/n)$ | Interaction of $Occ_{t-1}$ and $Cos(2\pi t/n)$ | ✓ | ✓ |
| 8 | NAOI | Northern Atlantic Oscillation Index | ✓ | ✓ |

'$n$' is 365 (366 in case of leap year)

covariates for the amount model is

$$X_A(s,t) = (1, Occ_{t-1}, Cos\left(\frac{2\pi t}{n}\right), Sin\left(\frac{2\pi t}{n}\right), Cos\left(\frac{4\pi t}{n}\right), Sin\left(\frac{4\pi t}{n}\right), Cos\left(\frac{4\pi t}{n}\right) * Sin\left(\frac{4\pi t}{n}\right),$$
$$Occ_{t-1} * Cos\left(\frac{2\pi t}{n}\right), NAOI) \tag{8}$$

The correlations for precipitation amount in the model are computed only for days when the precipitation was observed.

### 3.3 Model evaluation strategy

Although the model produces daily fields of precipitation, before evaluating the model for gridded data, we first evaluate at the individual locations where the observations are available. This is a common practice for validation of WGs that the input statistics must be reproduced. From the simulated gridded data, 30 years time-series of daily precipitation at the nearest grid point to the observation locations is extracted from each of the N=30 realisations. The mean of the simulated statistics in each realisation is compared with the observed statistics. The validation is carried out for daily and long-term statistics along with

considering more difficult statistics to be reproduced by the model. For the illustration of the results at individual locations, out of 29 stations, three example stations are selected which are: i) Oetz, ii) Pitztal Glacier, and iii) Prutz. These three stations are highlighted by red colour in the study area (Figure 1). The three stations are selected such that Oetz represents the results at valley stations, Pitztal Glacier represents the results at the high-mountain stations and the third station Prutz is one of the stations with different climatic characteristics and has climatic characteristics most distinct from those at the surrounding stations (and

is, therefore, most challenging). Note that Pitztal Glacier is the highest station amongst the 29 observation stations (see Table 1).



In the next step, we evaluate the model for its ability to reproduce spatial statistics. For that, gridded observed data are required. We use the Alpine Precipitation Grid Dataset (APGD) (Isotta et al. (2014)) from the Swiss Federal Office for Meteorology and Climatology (MeteoSwiss) which has a 5 km spatial and daily temporal resolution. These data are available from 1971 to 2008.

To obtain 30 years of gridded observations, we select the period from 1979 to 2008 for the validation of the simulated gridded data.

For the uncertainty estimation in the N=30 realisations, we use a tolerance interval (TI) (Patel (1986), Krishnamoorthy and Mathew (2009) and Young (2010)), instead of the conventional way of using a confidence interval for the sampling error. The

TI, along with the sampling error, provides bounds on observations from a population. Here, we use a parametric two-sided TI with a normal distribution. The TIs are computed for each of the statistics considered in this study, obtained from the simulated 30 realisations at each station. As uncertainty criteria, we select a confidence interval of 95% and a 99% proportion of the population for the TI. The TIs are shown in each figure as a shaded area around the curve and denoted as $TI_{99}^{95}$ throughout the article.

To quantify the model performance, along with various error metrics, we also take into account correlation coefficients (CC) and coefficients of determination ($R^2$). All together, we employ the following metrics: i) mean bias error (MBE), ii) mean absolute error (MAE), iii) root mean square error (RMSE), iv) CC, and v) $R^2$.

## 4 Results

An extensive evaluation of the model generated data is carried out here.

### 4.1 Evaluation at individual stations

#### 4.1.1 Daily occurrence probabilities at individual stations

First, we assess the performance of the occurrence model for the daily conditional probabilities on which the model is trained. These are important statistics as they are associated with the ability of the model to reproduce dry and wet spells. Figure 2

illustrates the annual cycle of the empirical and simulated daily conditional probability of a dry day following a dry day ($P_{dd}$) at the three selected stations, whereas Figure 3 illustrates the daily conditional probability of a wet day following a wet day ($P_{ww}$). The solid lines are the fitted curve using the Locally Weigthed Scatterplot Smoothing (LOESS) method (Cleveland (1979)) to the observed and simulated probabilities, respectively. The annual cycle of $P_{dd}$ is accurately simulated at Oetz and Pitztal Glacier while at Prutz, the model largely underestimates throughout the year. However, the seasonality in $P_{dd}$ is well

captured at Prutz. The annual cycle of $P_{ww}$ is well captured at Oetz and Pitztal Glacier, but slightly underestimated at Oetz during the entire year whereas at Pitztal Glacier, the model accurately reproduces the probabilities throughout the year.

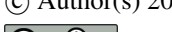



At Prutz, the model performs badly for $P_{ww}$ (Figure 3 (c)). The observed seasonality in $P_{ww}$ at Prutz is completely different compared to the seasonality at other stations which is not reproduced by the model at all. Similar to Prutz, the other two stations
(St. Leonhard im Pitztal-2 and St. Martin), which have very different climatic characteristics, also exhibit marked differences (not shown) between the simulated and the observed daily values of $P_{dd}$ and $P_{ww}$. What is noteworthy here is that the magnitude and the seasonality in both $P_{dd}$ and $P_{ww}$ at Prutz are close to the magnitude and seasonality of $P_{dd}$ and $P_{ww}$ at the valley station (Oetz). Similar behaviour is observed at the other two stations with poor performance. This is partly due to the fact that the covariates, which were selected at the majority of stations as optimal, did not optimally represent the statistics at Prutz
(and neither at the other stations with distinct climatologies). This indicates that the selected set of covariates is not able to reproduce seasonality at all the stations because a small subset of stations has distinctly different seasonality. Furthermore, the performance of the model at Prutz is influenced by the climate characteristics at the nearby stations with regard to magnitude and seasonality and not the other way around. At all other stations, $P_{dd}$ and $P_{ww}$ are well simulated. This suggests that the selected harmonics are capable to capture the seasonality in daily occurrence probabilities. Moreover, the temporal dependency
in the occurrence is well reproduced by the covariate $Occ_{t-1}$. In general, we find that the performance of the model at valley stations is similar to that at 'Oetz', and at the high-mountain stations similar to Pitztal Glacier.

To further examine the ability of the model to reproduce the observed climatology of wet days, we next consider the unconditional daily occurrence probability of wet days ($P_w$) (Figure 4). Again, at both Oetz and Pitztal Glacier, $P_w$ is very well
simulated and the annual cycle of $P_w$ is well captured by the model. At Prutz, the generator not only largely overestimates the probabilities but is also not able to accurately capture the seasonality. Again, both the seasonality and magnitude at Prutz in simulated probabilities are closer to those at valley stations (such as Oetz).

The performance metrics for daily occurrence probabilities are shown in Figure 8 (a). The metrics are computed for each of
the 29 stations and plotted as a boxplot. All the error metrics, CC and $R^2$ suggest the best performance is for $P_{dd}$ followed by $P_w$ and then $P_{ww}$. The high values of CC and $R^2$ for $P_{dd}$ and $P_w$ demonstrate the overall very good performance for these two statistics. Conversely, the small values of CC and $R^2$ for $P_{ww}$ reveal the relatively poor performance of the model. This is – at least partly – due to the fact that the model performs very poorly in generating precipitation series at the stations with distinct climatologies (see Figure 3 (c)).






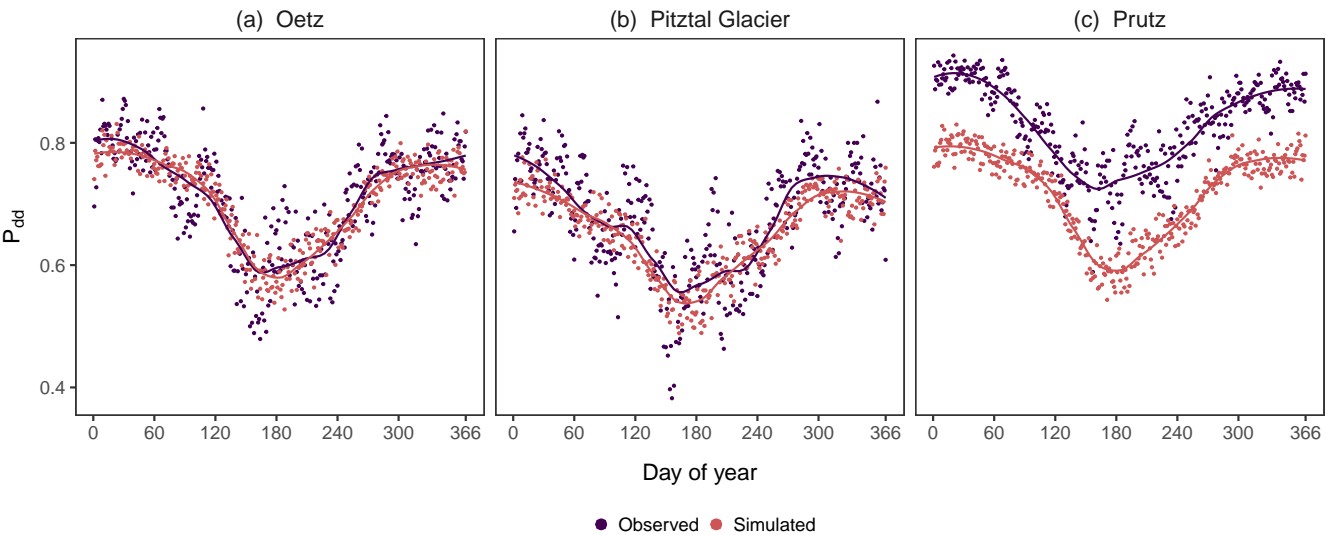

**Figure 2.** Daily conditional probability of a dry day following a dry ($P_{dd}$) day at three selected stations: (a) Oetz, (b) Pitztal Glacier, and (c) Prutz, (Figure 1 and Table 1). The observed probabilities are obtained from the observed 30 years (1981–2010). The simulated probabilities are the mean of the 30 realizations. The solid lines are the fitted curves using the LOESS method to the observed and simulated probabilities, respectively.

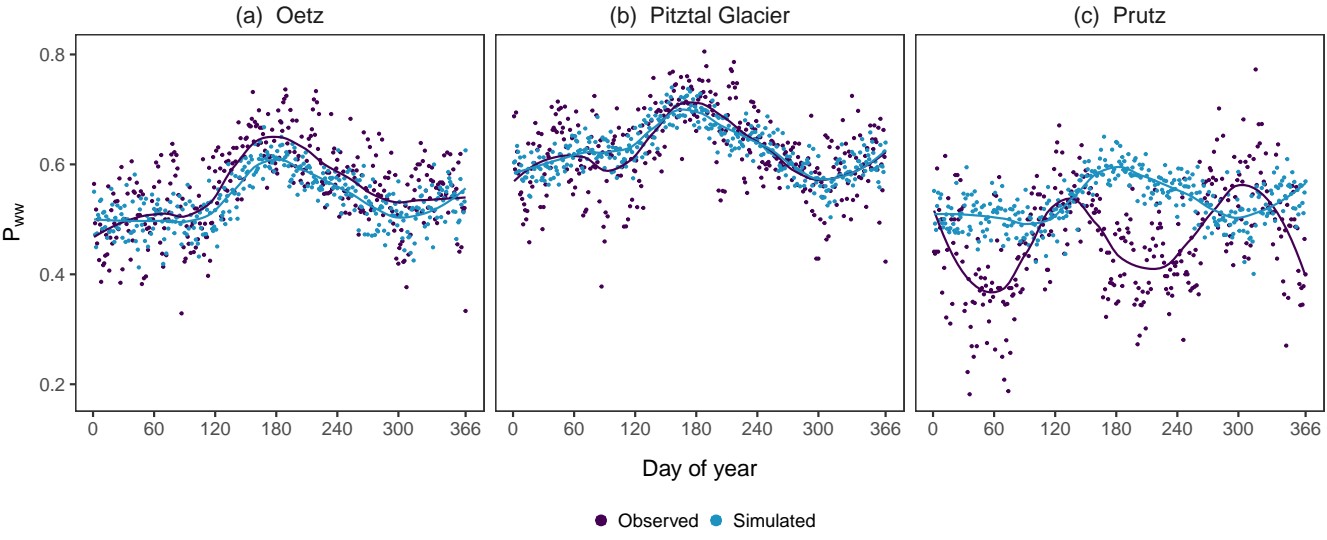

**Figure 3.** The same as Figure 2, but for daily conditional probability of a wet day following a wet day ($P_{ww}$).





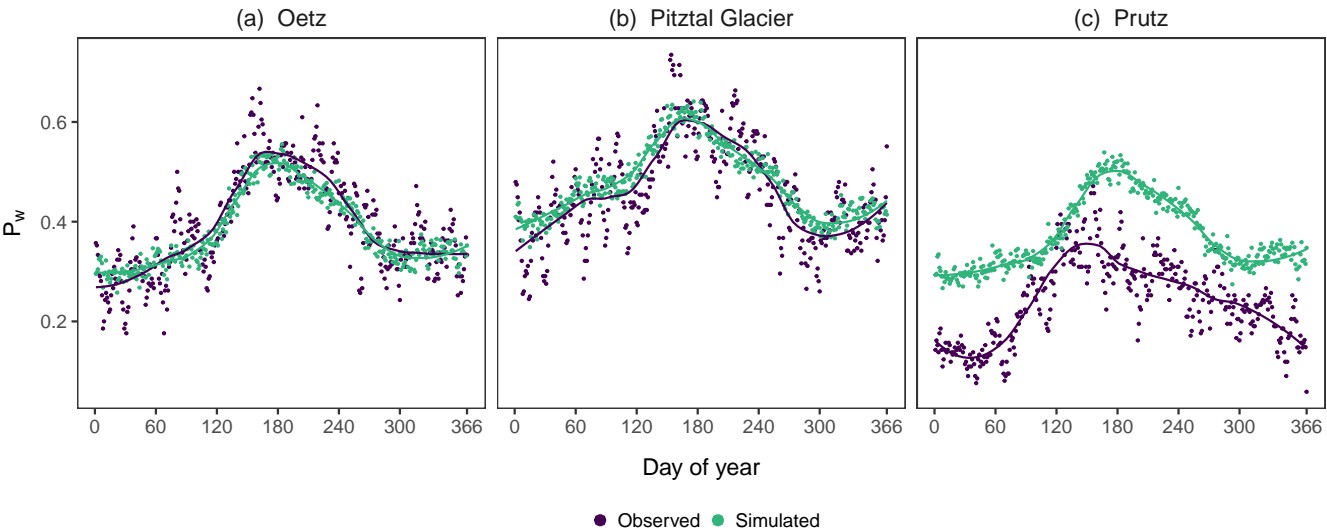

**Figure 4.** The same as Figure 2, but for daily occurrence probability of wet days ($P_w$).

### 4.1.2 Frequency of spells of different lengths at individual stations

Another important feature of the model is its ability to simulate long sequences of wet and dry days. Here, we examine the ability of the model to simulate wet and dry spells of different length at individual stations. Figure 5 displays the frequency of wet spells at the selected stations for the length 2 to 15 days. The precipitation generator is able to reproduce the wet spells of

different length at the valley and high-mountain station accurately. The shaded region is the tolerance interval $\text{TI}_{99}^{95}$ in the 30 realisations. The observed values of the spells of different length are within the tolerance interval which shows that the model does an excellent job in reproducing even the longer spells. It is noteworthy that the model captures the spells very accurately even if it is not trained on these statistics. There are very few occurrences of wet spells longer than 10 days at the majority of stations in the observed data which are also well reproduced. At Prutz, the model overestimates the wet spells of all different

lengths. The observed values of the spells are not within the $\text{TI}_{99}^{95}$ which shows that the overestimation is consistent in all the 30 realisations. The overestimation occurs because the model is not able to reproduce the conditional probabilities $P_{ww}$ at Prutz even reasonably well. In fact, the large overestimation in $P_w$ at Prutz (Figure 4) ultimately contributes to the overestimation in wet spells.

Figure 6 demonstrates the frequency of dry spells of length 2 to 25 days at three selected stations. The model is able to simulate the dry spells of all different length at the valley station as well as high-mountain station with great accuracy. There are very few occurrences of extreme spells of length longer than 15 days in the observed data which are also well reproduced by the model. At Prutz, the model overestimates the spells of shorter length of up to 4 days but does a very good job for the longer spells. Again, the shaded regions are the statistical tolerance level $\text{TI}_{99}^{95}$ in the 30 ensembles. The observed values of the spells





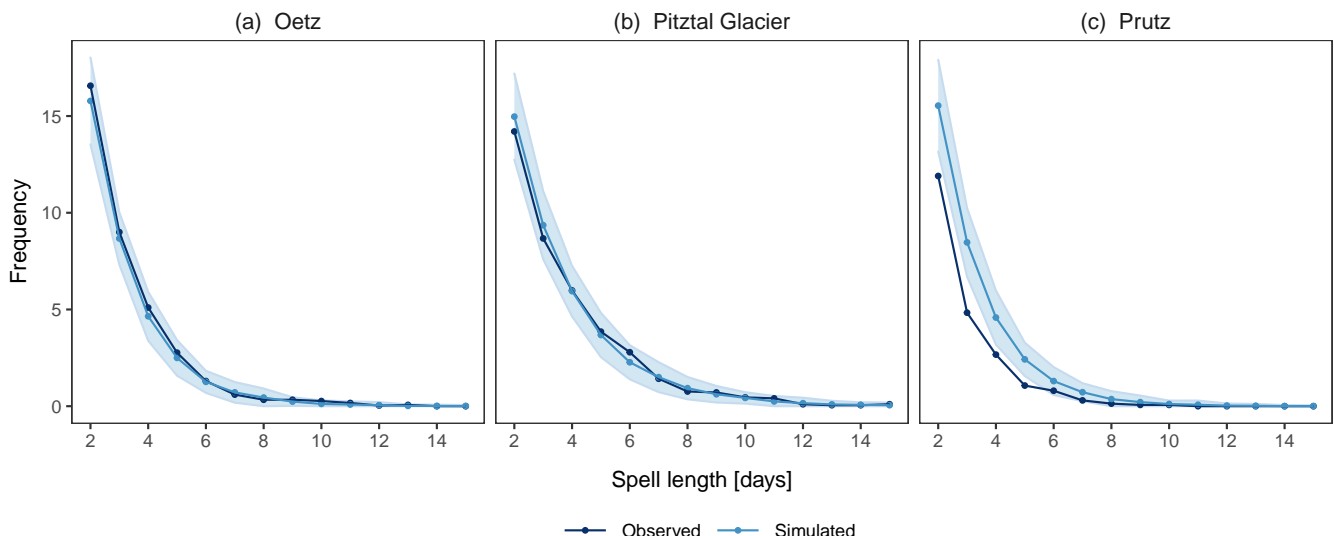

**Figure 5.** Frequency of wet spells of different legnth at three selected stations: (a) Oetz, (b) Pitztal Glacier, and (c) Prutz. The frequency of the observed spells is obtained from the observed 30 years (1981–2010). The frequency of the simulated spells is determined for each of the 30-year simulations separately and averaged over the 30 realizations. The shaded region is the selected tolerance interval $TI_{99}^{95}$ for the simulated values in the 30 realisations. The two-sided tolerance interval is specified for 99% of the population and with 95% confidence level.

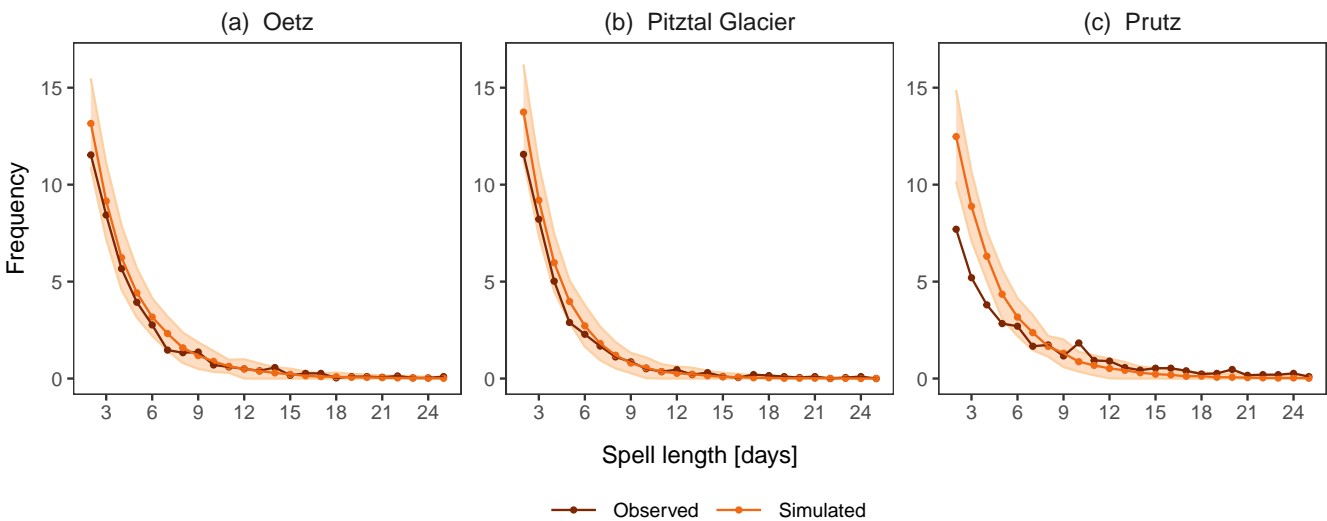

**Figure 6.** As Figure 5, but for the frequency of dry spells of different legnth.





are within the tolerance region except for the shorter spells even at Prutz. The worst performance for both wet and dry spells has been observed at Prutz. In general, for the stations in the valley, the model does a better job than at the high-mountain stations (figure not shown). The previous day's occurrence as a covariate ($Occ_{t-1}$) is able to simulate the long sequences of both wet and dry days very well at the majority of the stations with a great accuracy. The occurrence model satisfactorily reproduces the occurrence pattern at the majority of the stations, except at the few distinct stations with peculiar climatologies.


As for dry and wet spells, the model performance is mostly similar for both the spells (Figure 8 (b)) except that the RMSE is, in general, larger for wet spells suggesting overall slightly worse performance for wet spells. The CC and $R^2$ are nearly 1 which indicates an excellent agreement of the model with the observations for both dry and wet spells.

### 4.1.3 Monthly mean precipitation at individual stations

An important aspect of the precipitation generator is its ability to reproduce the amount of precipitation observed at the stations. As the model for amount is the gamma distribution at the observed locations, the mean, which is the product of the shape and scale parameters of the gamma distribution, should be well reproduced. Figure 7 displays the monthly mean of precipitation in the observed and simulated data at the selected stations. At both Oetz and Pitztal Glacier, the model is able to reproduce the mean very well as the observed values are within the $\mathrm{TI}_{99}^{95}$ (the shaded region). At Prutz, the model underestimates the mean

in April, May, October and November while in other months, the model is able to reproduce the mean reasonably well as the observed values are within the $\mathrm{TI}_{99}^{95}$. The performance metrics are displayed in Figure 8 (b). For the monthly mean precipitation, the model performs very good which can be seen by the small magnitudes of the errors – typically less than 0.5 mm/day, and strong values of CC and $R^2$.






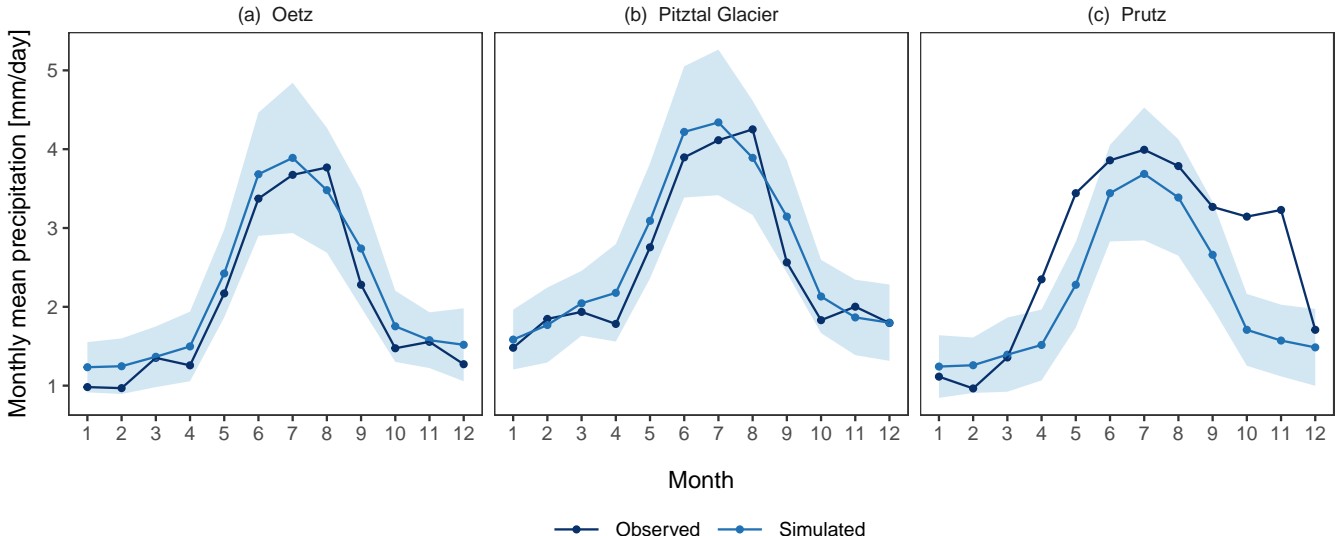

**Figure 7.** Monthly mean precipitation [mm/day] at the three selected stations: (a) Oetz, (b) Pitztal Glacier, and (c) Prutz. The observed values are obtained from the observed 30 years (1981–2010) and the simulated values are the mean of the 30 realisations. The shaded region is the selected tolerance interval $\mathrm{TI}_{99}^{95}$ for the simulated 30 realisations.

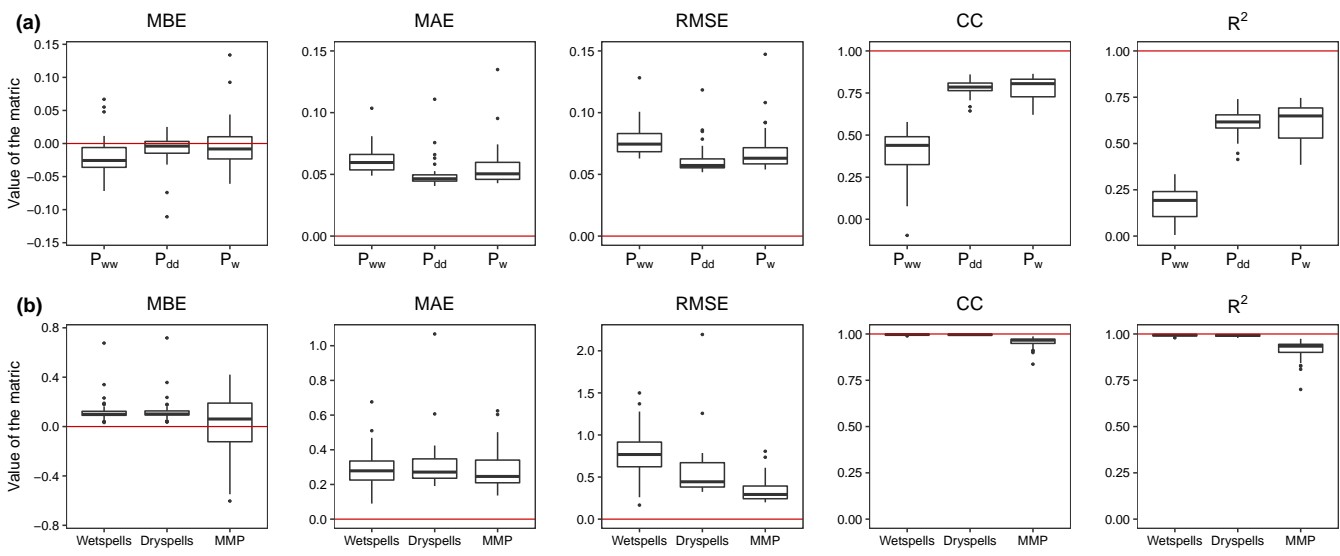

**Figure 8.** Performance metrics for the model at individual stations: (a) for daily occurrence probabilities ($P_{dd}$, $P_{ww}$ and $P_w$), and (b) for the frequency of dry spells and wet spells of different length and monthly mean precipitation (MMP). The unit for the MBE, MAE and RMSE for dry/wet spells are [per year] and for MMP is [mm/day]. The red horizontal line in each panel plot corresponds to the optimal performance for the corresponding metric.



**Figure 9.** QQ plot of daily preipitation at each of the 29 individual stations with observations (Table 1). The shaded region is the selected two-sided tolerance interval $TI_{99}^{95}$ for the simulated 30 realisations. The black solid line in each panel plot corresponds to the 1:1 line. Note that the scaling on the axes is not the same for all the stations.

### 4.1.4   QQ plot at individual stations

Here, we examine the QQ plot at each individual station where the observed data are available (Figure 9). It can be seen that the distribution of the precipitation is accurately simulated at a majority of the stations. Largest discrepancy between the observed and simulated distribution is found at Prutz which has the largest spread in the observed data and that is not well reproduced by the model. Apart from that, there are some stations including Dresdner Huette where the higher quantiles are not well reproduced. These are the stations which have longer tails in the distribution of precipitation in the observed data. This is a commonly reported problem in the literature that the gamma distribution is not adequate to simulate extremes.



We further examine the distributions of the generated data for each month at each of the 29 stations using the Kolmogorov-
Smirnov test and the Wilkoxon-Mann-Whitney test. The results are shown in the Supplement (Figure S1–S2). As revealed by
the QQ plots, the worst performance is observed at Prutz, St. Martin and St. Leonhard im Pitztal-2 which have the distinct
climatologies.

## 4.2   Evaluation of simulated gridded data

### 400   4.2.1   Frequency of areal spells of different length

One of the most challenging features of the gridded weather generator is its ability to reproduce the areal spells of wet and
dry days of different length. This is one of the sought-after features in agricultural and hydrological applications. We define an
areal wet (dry) spell as the number of consecutive days when 95% of the study area is wet (dry). Figure 10 illustrates the areal
spells of dry and wet days of different length in the observed and simulated data. It can be seen that the areal dry spells are
better simulated than the areal wet spells. The areal dry spells of length greater than 2 days are accurately simulated, whereas
the areal dry spells of 2 days are overestimated. For areal wet spells, the model underestimates the spells of all length. Larger
discrepancies are found for shorter spells.

For the areal spells, the performance metrics are shown in Table 3. For areal dry spells, the error statistics suggest that the
model has a tendency to overestimate the spells. This is because the model largely overestimates the areal dry spells of shorter
lengths. While the CC is perfect and $R^2$ is also nearly 1 which suggest an excellent performance of the model. For areal wet
spells, the error metrics MBE, MAE and RMSE are small where a negative value of MBE indicates overall underestimation in
the wet spells and strong CC and $R^2$ suggest a very good agreement with the observed values.

**Table 3.** Performance metrics for the gridded model for reproducing areal statistics

| Statistics | MBE | MAE | RMSE | CC | $R^2$ |
| --- | --- | --- | --- | --- | --- |
| Frequency of areal dry spells of different length | 1.02 [per year] | 1.38 [per year] | 3.34 [per year] | 1.00 | 0.99 |
| Frequency of areal wet spells of different length | -0.41 [per year] | 0.62 [per year] | 1.30 [per year] | 0.99 | 0.97 |
| Monthly mean areal precipitation | -0.23 [mm/day] | 0.26 [mm/day] | 0.33 [mm/day] | 0.96 | 0.93 |

### 415   4.2.2   Spatial distribution of occurrence probabilities

The spatial distribution of the simulated probabilities of wet days ($P_w$) for each month (Figure 11) is compared to that of
the (gridded) observations (Figure 12). It is noteworthy that the model generated data have 1 km spatial resolution, while the
observed gridded data have 5 km resolution. The model is able to generate the seasonality in the spatial distribution of $P_w$
very well (Figure 11). In particular, the higher probabilities in spring and summer months are well reproduced. Also, at 1 km



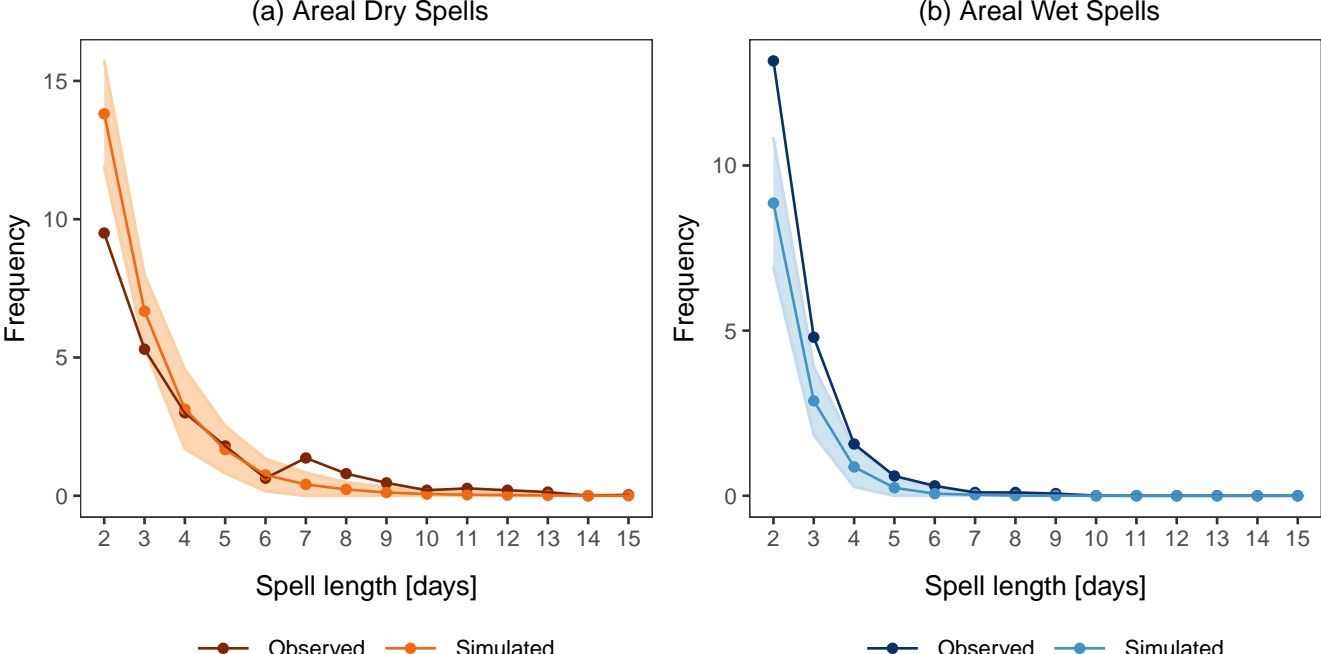

**Figure 10.** Frequency of the areal spells of different length (see Section 4.2.1): (a) Dry spells, (b) Wet spells. The observed spells are obtained from the APGD gridded data of 30 years (1979–2008). The simulated spells are the mean of the 30 realizations. The shaded region is the selected tolerance interval $TI_{99}^{95}$ for the simulated data. The two-sided tolerance interval is specified for 99% of the population and with 95% confidence level.

high resolution, the influence of topography on the spatial distribution of $P_w$ is clearly visible. This is due to the inclusion of elevation in the kriging interpolation.

To test the model performance at each grid point, we upscale the simulated data from 1 km to 5 km and the difference between the observed and the simulated data is shown in Figure 13. It can be seen that in spring and summer the biases are small and

mostly similar throughout the domain, while in winter there is an overestimation in the south-eastern part of the region and a slight underestimation in the north-western part. This is because in the south-eastern part of the study area the density of stations is very low. There is one station (St. Martin) that is heavily influencing that area which has the highest probability of precipitation occurrence and that is manifested through the the spatial interpolation of the model parameters through kriging (see Section 2) to generate the gridded fields. Except for summer, this station has a much higher probability of precipitation

occurrence relative to the other stations. This is reflected in the simulation also that in summer and mainly in August in the south-eastern part the bias is mostly zero.



**Figure 11.** Spatial distribution of occurrence probability ($P_w$) in each month in the simulated data. The probabilities for each month are computed at each grid point in each of the 30 realisations and the mean of the 30 realisations is shown. The spatial resolution for simulated data is 1×1 km.





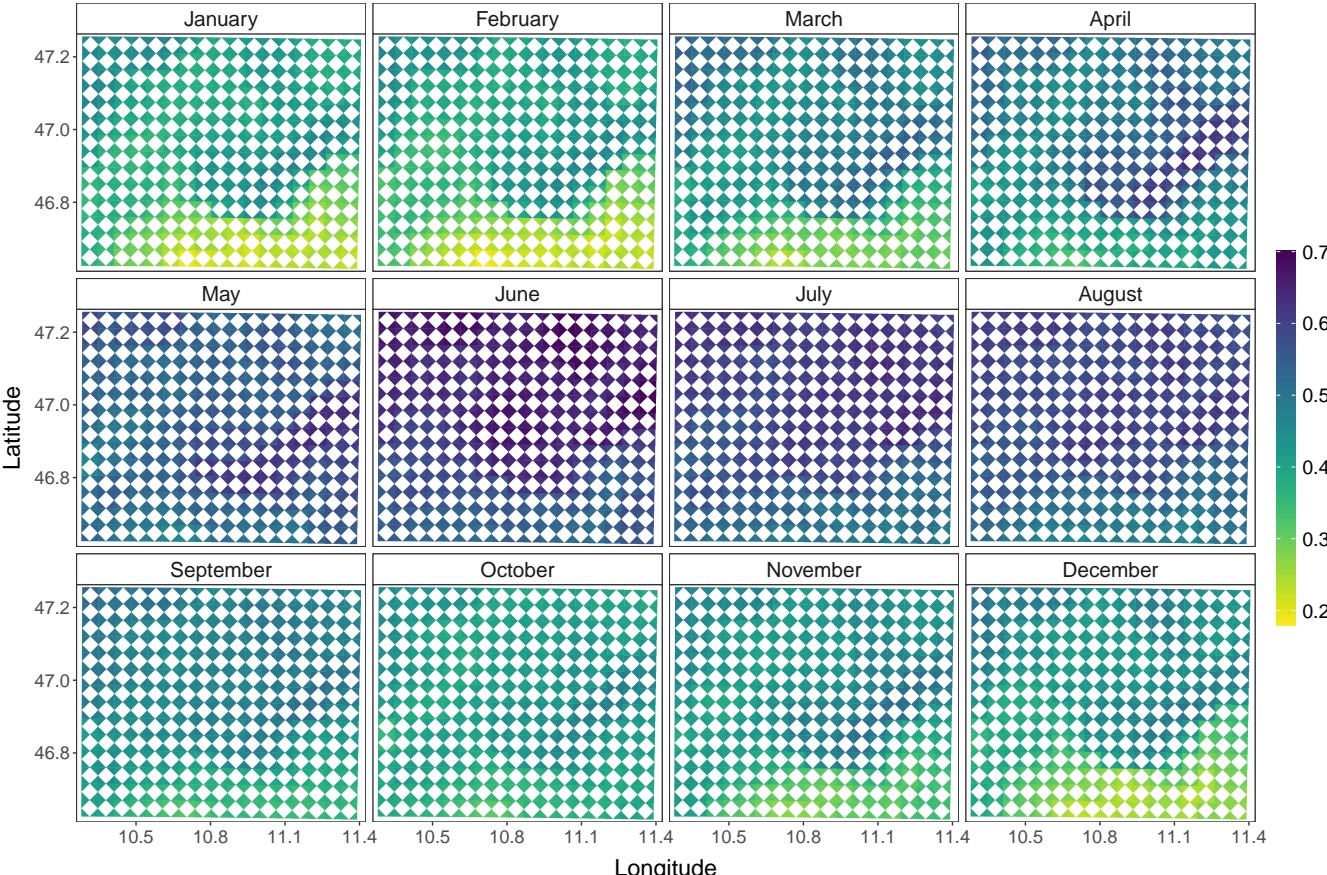

**Figure 12.** Spatial distribution of occurrence probability ($P_w$) in each month in the 30 years (1979–2008) APGD observed gridded data. The spatial resolution for APGD data is 5×5 km.

Figure 17 (a) illustrates the performance metrics for the spatial distribution of $P_w$ for each month. All the three error statistics are within 0.1 which suggest a very good performance of the model. The largest errors are for August where negative MBE indicates overall underestimation in $P_w$, while the largest overestimation is found in February. The lowest overall bias is found in September. The highest value of CC and $R^2$ is observed in May. In colder months, i.e. in winter and autumn the correlations between the observed and simulated probabilities are weak and also the $R^2$ is small which suggests poor performance for these months. The worst performance is observed in January where the CC and $R^2$ are almost zero, while the best is found in May.

### 4.2.3 Spatial distribution of mean wet-day daily precipitation

Next, we consider the mean wet-day daily precipitation for assessing the ability of the model to reproduce the observed climatology of precipitation amount over the region. We consider the mean of daily precipitation on wet days in each month over the





**Figure 13.** Bias (Simulated-Observed) in spatial distribution of occurrence probability ($P_w$) in each month. Note that the simulated data has been upscaled to the resolution (5×5 km) of the AGPD data set.

30 years of simulated 30 ensembles, i.e. 900 years of data at each grid point (Figure 14) and compare it with the observations (Figure 15). Figure 16 depicts the bias in the simulated and observed values at each grid point. The model is able to simulate
the spatial seasonal variability in precipitation very well. In particular, the summer precipitation due to the convective processes and thunderstorms, which account for high amounts of precipitation, is well reproduced. In the colder months from October to January, the precipitation amount in the simulated data has almost no elevation dependency and hence the generator has a tendency to generate a more uniform field of daily precipitation amount (Figure 14) across the region. In contrast with the largely overestimated spatial probability of occurrence in the south-eastern part of the study area, the model underestimates the
precipitation amount. St. Martin does not only have a high frequency of wet days but also the highest precipitation amount in all the months among the 29 stations. The gamma distribution is not able to reproduce the large amount at this station which



**Figure 14.** Spatial distribution of simulated mean wet-day daily precipitation for each month [mm/day]. The mean wet-day daily precipitation is obtained at each grid point for each month in each of the 30 realisations and the average of the 30 realisations is shown. The spatial resolution for simulated data is $1 \times 1$ km.

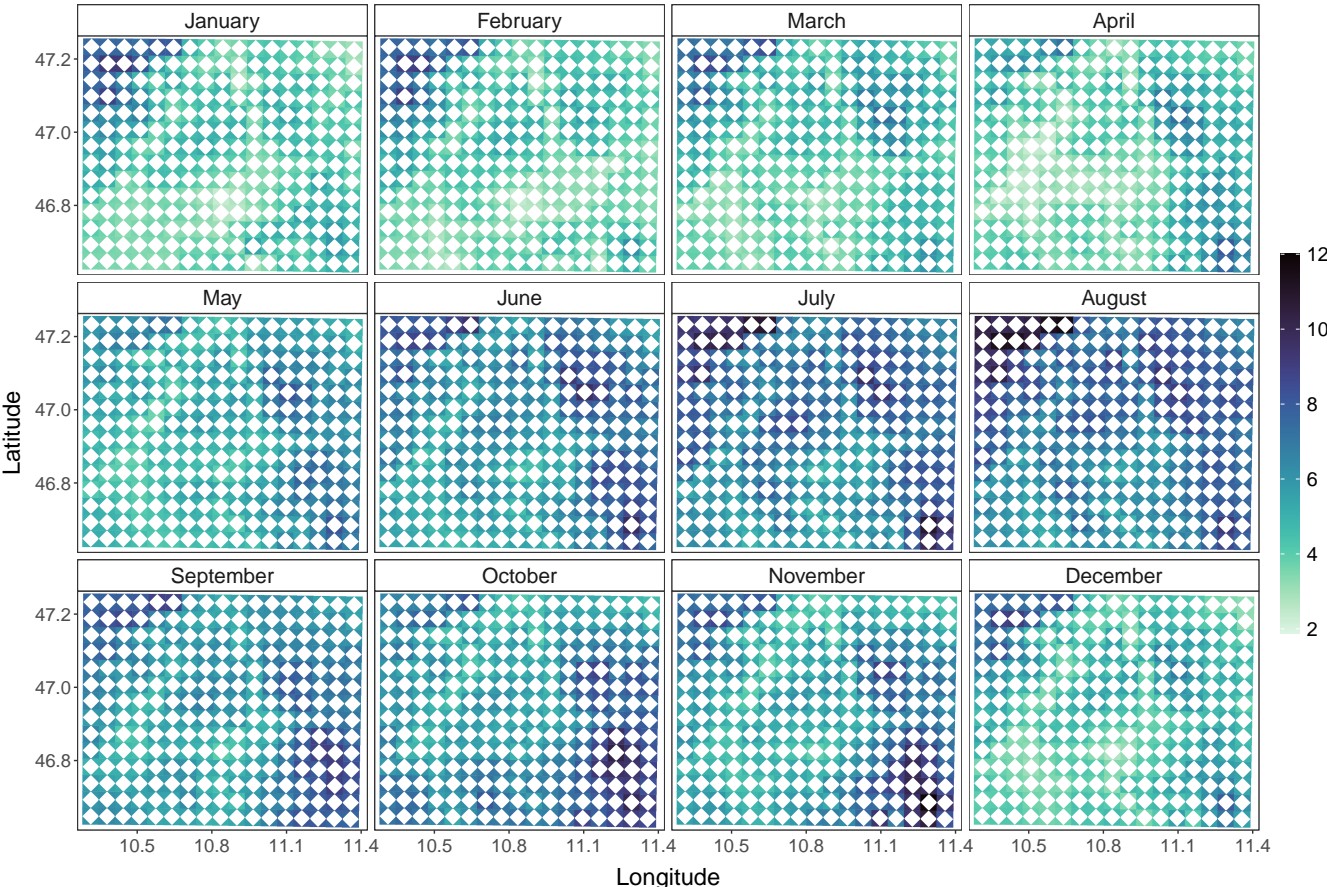

**Figure 15.** Spatial distribution of mean wet-day daily precipitation for each month [mm/day] in the APGD observed 30 years (1979–2008) gridded data. The spatial resolution for APGD data is 5×5 km.

results in underestimating the amount in the surrounding area also.

Another important aspect to notice is that in the observed data APGD, the probabilities of wet days in the south-eastern part of
the region in the colder months is low (Figure 12), whereas in the same part of the region, the precipitation amount is high –
particularly in October and November (Figure 15). The model fails to capture this behaviour. This is also because in this part
of the study area the model has to extrapolate the parameters as there is no observed station present beyond St. Martin. Hofstra
et al. (2010) found that the density of station network used for interpolation influences the distribution of precipitation and
the areal mean amount of precipitation. They found that when fewer stations are used, precipitation is over-smoothed which
leads to a strong tendency for interpolated values to be less than the "true" value, and the effect was the greatest for higher per-
centiles. Dresdner Huette also has a larger amount in observed data in autumn (and also in winter) compared to other stations.
The model underestimates the amount in the region surrounding this station also, which is mainly in October and November





**Figure 16.** Bias (Simulated-Observed) in the spaial distribution of mean wet-day daily precipitation [mm/day] for each month. Note that the simulated data has been upscaled to the resolution (5×5 km) of the AGPD data set.

(see the north-eastern part in October and November in Figure 16). Another reason for the underestimation in autumn could be the inability of the model to simulate orographic precipitation particularly related to föhn events. This altogether leads to large

underestimation in autumn in this region.

Figure 17 (b) depicts the performance metrics for the spatial distribution of mean wet-day daily precipitation for each month. The MBE, MAE and RMSE are the largest in November and negative value of MBE approximately -2 mm/day suggests large underestimation in that month. For October also, large underestimation is found, while the smallest error metrics are found in

March followed by February. The high values of CC and $R^2$ are in February and March which suggest the best performance for these months, while the worst performance is found in September. Contrary to the bad performance for spatial distribution of $P_w$, the spatial distribution of mean precipitation is better reproduced in winter (compare Figures 13 and 16).



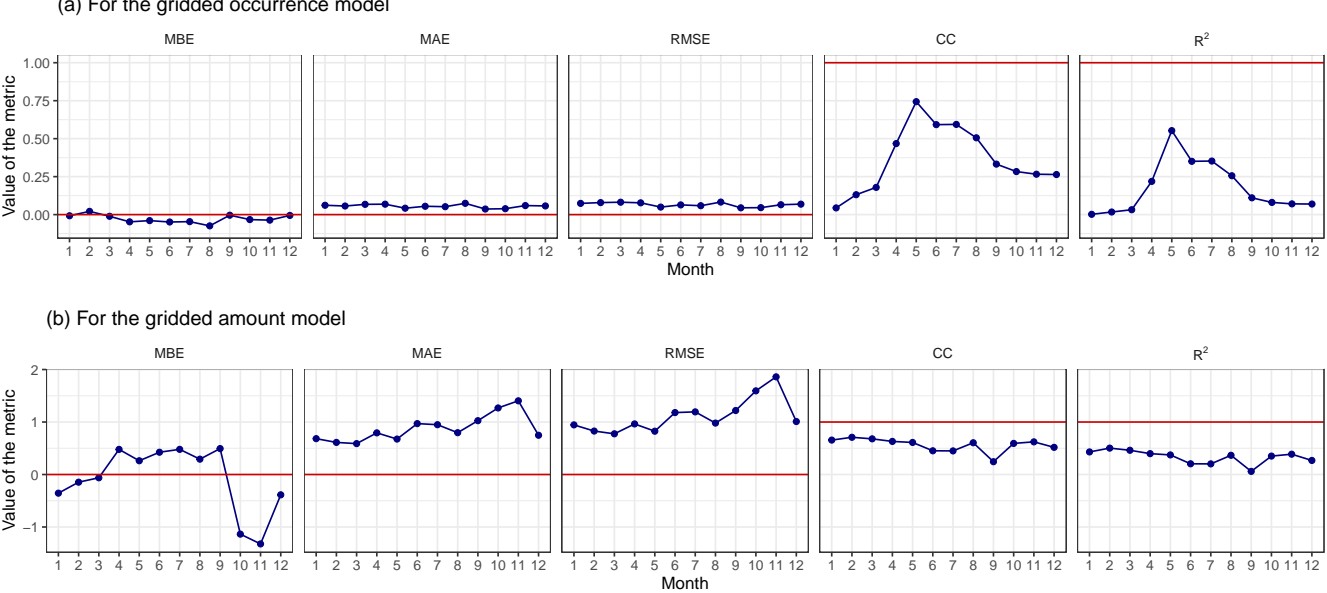

**Figure 17.** Performance metrics for the gridded model: (a) for spatial distribution of probabilities of wet days ($P_w$) in each month, and (b) for spatial distribution of mean wet-day daily precipitation where the unit for the MBE, MAE and RMSE is [mm/day]. The red horizontal line in each panel plot corresponds to the optimal performance for the corresponding metric.

### 4.2.4   Monthly mean areal precipitation

Next, we assess the ability of the precipitation generator to provide an areal climatology of the precipitation amount. This is also one of the desired characteristics for impact modelling. Figure 18 displays the areal precipitation mean for each month in the observed as well as simulated gridded data. The precipitation generator, except in autumn, simulates the mean areal precipitation in all seasons with good accuracy as the observed values are within the $TI_{99}^{95}$. As seen in the spatial distribution of the amount of precipitation (Figure 14), in autumn the model underestimates mainly in October and November. The statistics

are shown in Table 3. The small negative value of MBE indicates overall slight underestimation, while MAE and RMSE are also small about less than 0.4 mm/day. The high values of CC and $R^2$ show that the model estimated means are in very good agreement with the observed.

### 4.3   Comparison between anisotropic and isotropic model using KED and OK

Here, we compare the results of our simulations i.e. using KED in the anisotropic model denoted Aniso-KED with three different model set-ups: i) by considering OK in the interpolation of the parameters of the anisotropic model (Aniso-OK), ii) using the original isotropic model which uses OK for the interpolation of the parameters (Iso-OK), and iii) using the isotropic model



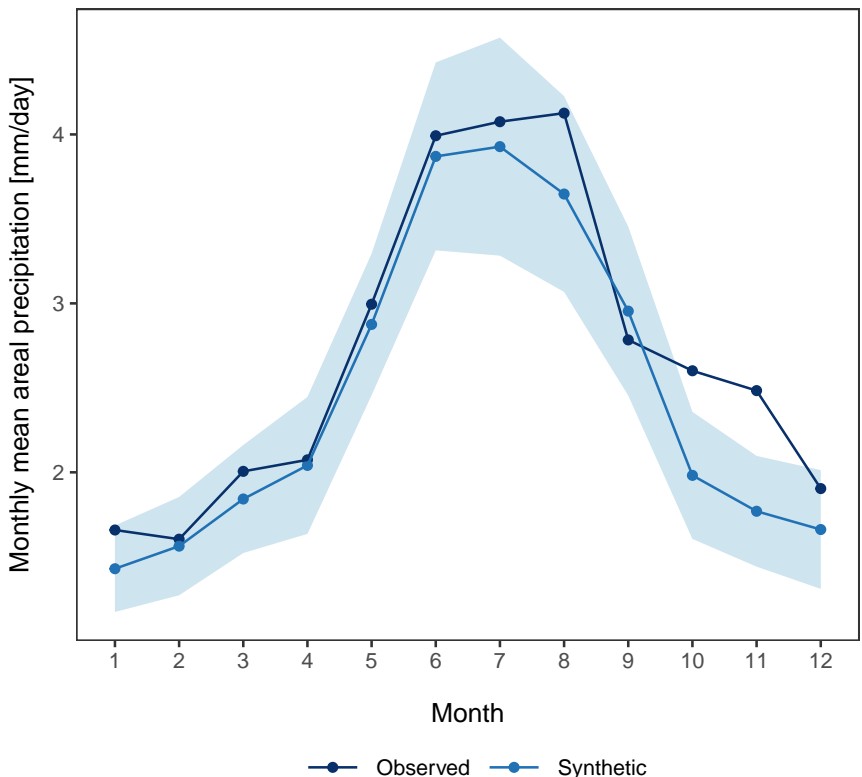

**Figure 18.** Monthly mean areal precipitation [mm/day]. The observed values are obtained from the APGD observed 30 years (1979–2008) and the simulated values are the mean of the 30 realisations. The shaded region is the selected two-sided tolerance interval $TI_{99}^{95}$ for the simulated data.

with KED (Iso-KED). We examine the results for the monthly sum of areal mean daily precipitation in these four cases of
simulations (experiments) with the observed gridded data APGD. Figure 19 displays the distributions of monthly sum of areal mean daily precipitation for the simulated 900 years for each of the four experiments and the observed 30 years gridded data.

The model performance for Aniso-KED and Iso-KED is mostly similar in all the months, whereas the model performance using Iso-OK and Aniso-OK is similar in all the months. The model performance varies greatly from month to month for all
four experiments, but both the experiments involving KED (Aniso-KED and Iso-KED) outperform those using OK in all the months. It is evident that by allowing the elevation as a covariate in the kriging interpolation for prediction at each grid point, the amount of precipitation is considerably improved.





The median for both the KED experiments is overestimated in the months February, March, May and September and in the
rest of the months, the median is underestimated. The inter quartile range (IQR) in Figure 19 shows the inter-annual variability
(IAV) but the figure also shows the intra-seasonal and inter-seasonal variability. The model reproduces the intra-seasonal and
inter-seasonal variability very well in all the four experiments but in general better for the experiments when KED is employed.
The IAV is better simulated in summer than in colder months. The best performance is found in July, when the variability is
larger in the simulated data for all the four experimental set-ups compared to the observed data. For both the simulations using
KED, the median in July is furthermore very close to the observed value. This is a remarkably good performance as WGs are
typically criticized to have a tendency to underestimate the low frequency variability, as it is the case for most of the months
in our precipitation generator. Conversely, our model – for all the four experiments – is not able to reproduce the larger IAVs
in other months (particularly October). This also could be one of the reasons for the large underestimation in the precipitation
amount over the study area in the autumn season as discussed in the previous sections. It is noteworthy that from October to
December, the model performance is essentially similar in all the four experiments. This shows that regardless of the type of
correlation structure and the interpolation method, the model is unable to capture the spatial distribution of precipitation in
those months. In general, the model performance is better in warmer months than in colder months.

The differences in the observed and simulated median and IQR in each month for each of the four experiments are listed in
Table 4. Overall, the experiments with KED outperform those using ordinary kriging and the Iso-KED combination is slightly
superior to the fully anisotropic combination. Apparently, even for such a small region in complex terrain as the present study
area, an isotropic covariance is adequate to reproduce the precipitation fields. In the observed data, anisotropy is indeed present
but the difference between the variation in the correlations with horizontal distance compared to that in the vertical is very
small. Hence, there is almost no difference in the performance of the model in isotropic and anisotropic formulation. The
two experiments with OK not just underestimate the amount of precipitation but also lack the topographical influence on the
simulated precipitation amount and rather produce smooth precipitation fields over the region (see Figure S3 in Supplement).
However, the influence of topography must be included in the model, to realistically simulate the precipitation fields, as we
showed here by considering KED interpolation.

As for the occurrence model, the covariance structure has a slight influence on the model performance. Figure 20 compares the
areal dry and wet spells for all the four experiments with those observed. The performance of the anisotropic model is better
for dry spells where Aniso-KED performs sightly better than Aniso-OK. Contrarily, the performance of the isotropic model is
better for wet spells where Iso-KED performs slightly better than Iso-OK. For both areal dry and wet spells, KED interpolation
adds little value to the simulation over using OK.





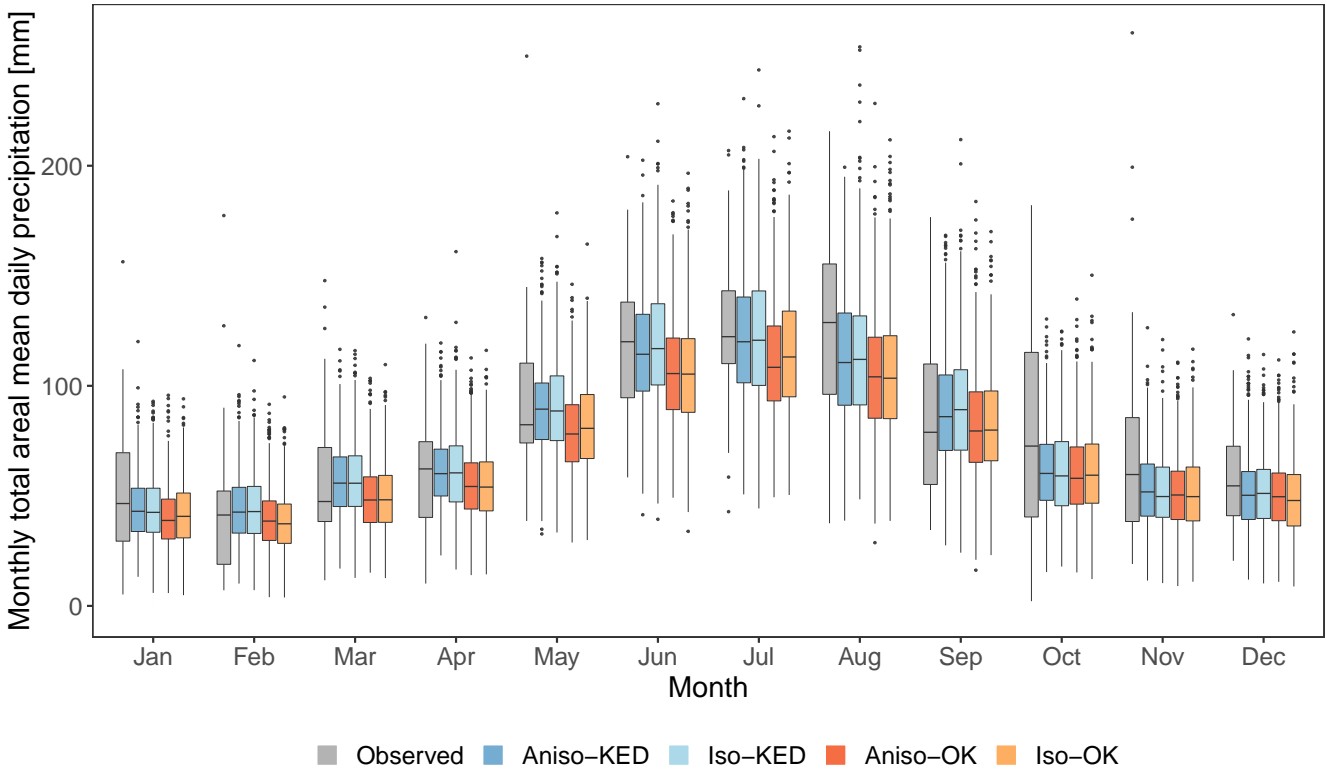

**Figure 19.** Monthly sum of areal mean daily precipitation [mm] for each month in APGD observed 30 years (1979–2008) and simulated 900 years (30 realisations of 30 years) of data for the four experiments: Aniso-KED, Iso-KED, Aniso-OK and Iso-OK (see text for specifications).

## 5    Discussion

The extended model proposed here for simulating precipitation adds substantial value over the original framework of Kleiber et al. (2012) to the simulation of gridded precipitation fields in the highly complex mountainous region in the Austrian Alps. The precipitation generator, using observed meteorological station data as input, is able to provide high resolution gridded

precipitation fields in complex terrain and thus provides data where historical observations are not available. As a statistical downscaling tool, this model does not require any large computational resources. Obviously, compared to a single-site WG, it requires additional computing resources, but it is still very parsimonious and fast compared to dynamical downscaling models. It can be easily run on any high-end personal computer.

In this study, we have tested two extensions of the original isotropic model to be used for applications in complex topography. First, to include anisotropy in the covariance function and second, to apply KED in the interpolation of the parameters instead of OK. The major improvement in the results from our model comes from the KED interpolation, rather than the included





**Table 4.** Difference (Simulated-Observed) in the median and interquartile range (IQR) of the observed and simulated monthly sum of areal mean daily precipitation [mm] in each month and each of the four experiments. The bold values indicate the best performance of the model in each month.

| Month | Aniso-KED | Iso-KED | Aniso-OK | Iso-OK |
|---|---|---|---|---|
| | **Difference in Median (IQR) [mm]** | | | |
| Jan | **-3.54** (-20.52) | -4.01 (-20.16) | -7.70 (-22.06) | -5.84 ( **-19.79**) |
| Feb | **1.33** (-12.52 ) | 1.57 ( **-11.93**) | -2.76 (-15.34) | -3.99 (-15.44) |
| Mar | 8.35 (-11.10) | 8.31 (**-10.66**) | **0.70** (-12.92) | 0.78 (-12.32) |
| Apr | -2.15 (-13.13) | **-1.80** ( **-8.94**) | -7.98 (-13.42 ) | -8.26 (-12.15 ) |
| May | 7.08 (-10.58) | 6.26 (**-6.86**) | -4.20 (-10.33) | **-1.63** (-7.19) |
| Jun | -5.69 (-8.56) | **-3.13** ( **-6.55**) | -14.47 (-10.95) | -14.71 (-9.98) |
| Jul | -2.30 (5.87) | **-1.61**(9.83) | -13.91 ( **0.95**) | -9.22 (5.84) |
| Aug | -18.18 (**-17.34**) | **-16.76** (-18.80) | -24.67 (-22.44) | -25.30 (-21.56) |
| Sep | 7.10 (-20.44) | 10.28 ( **-18.20**) | **0.60** (-22.71) | 1.00 (-23.03) |
| Oct | **-12.42** (-49.34) | -13.56 (**-45.64**) | -14.64 (-48.79) | -13.24 (-47.90) |
| Nov | **-7.90** (-23.54) | -10.00 (-24.40) | -9.30 (-25.25) | -10.04 ( **-22.74**) |
| Dec | -4.29 (-9.79) | **-3.43** (-9.26) | -4.94 (-9.89) | -6.65 (**-8.19**) |

anisotropy in the covariance structure. This suggests that there is no strong directional dependency in the precipitation simulation. Although there are minor differences in the model performance using the isotropic and anisotropic covariance functions,

it can be concluded that isotropic covariance function is sufficient even for small-scale topographic variability as in the present study in the European Alps. However, the topographical details must be included in the interpolation of the parameters of the model. Similar results can be expected for complex terrain in other mountainous regions.

At individual locations with observations, the model satisfactorily reproduces various observed statistics and the overall distribution of precipitation. The model is also able to capture spatial and temporal variability over the entire region reasonably

well. It is capable to simulate the dry day statistics over the whole region very well, while for the wet day statistics, there is an underestimation observed. The frequency of areal dry spells of one or two days is strongly overestimated. The model uses previous day's occurrence as a covariate, which at an individual location creates a first-order two-state Markov chain. Dabhi et al. (2021) applied a first-order two-state Markov chain model at stations covering different climate zones in Europe and

found that it has a tendency to overestimate dry spells. The first-order Markov chain uses occurrence from only one day in the past, there may, however, be longer lasting correlations present in the data. Considering the occurrence from two or more days in the past, i.e. forming a second-order or higher order Markov chain at individual locations may potentially improve the results for both dry and wet spells. For example, Wilson Kemsley et al. (2021) studied the order of Markov chain in different climate regimes across the world where they showed that the third-order model reproduces observed dry spell distributions the

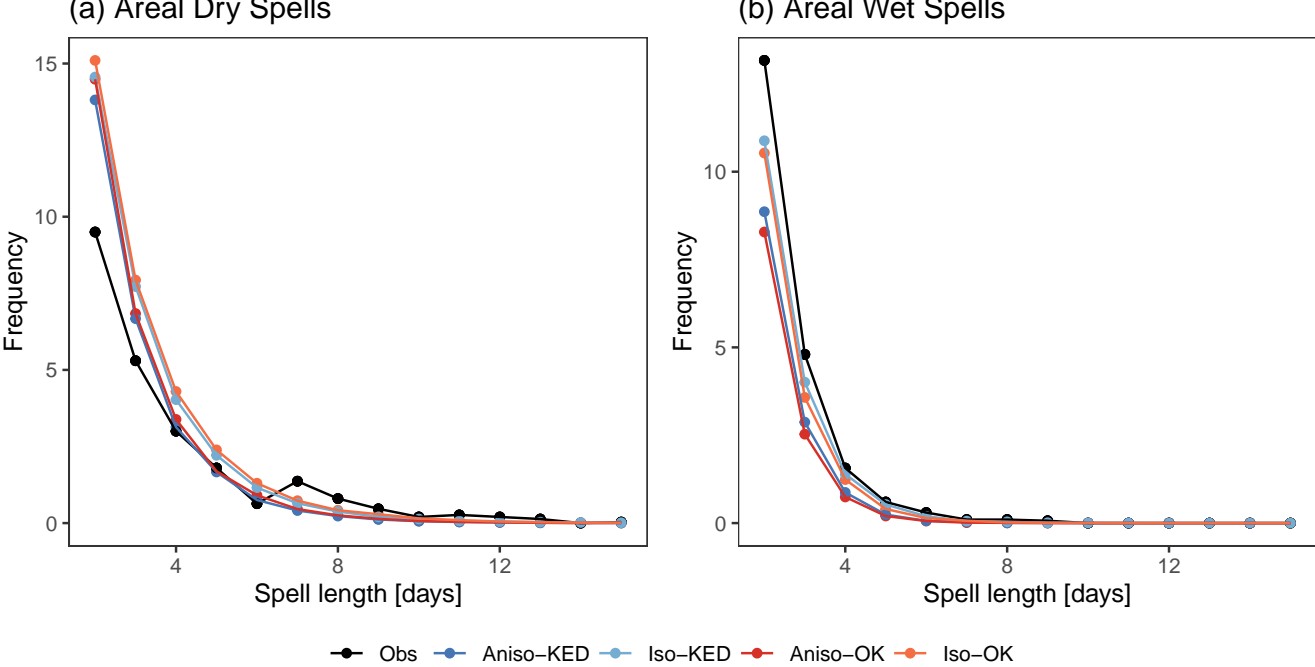

**Figure 20.** Frequency of areal spells [per year] of different length in each of the four experiments with 900 years of simulated data and in APGD observed 30 years (1979–2008) data: for (a) areal dry spells, and (b) areal wet spells. For the simulated dry and wet spells, their frequency is determined for each of the 30-year simulations separately and averaged over the 30 realizations.

best. Alternatively, by allowing other meteorological variables such as windspeed, humidity as covariates in the GLM can also improve the results (Ataharul Islam and Chowdhury (2006)).

The model captures the month-to-month variability in the monthly sum of precipitation very well which is due to the time-dependent harmonics of sine and cosine as covariates in the modeled spatial covariance structure. However, the inter-annual
variability is largely underestimated mainly for the colder months. Even if we adopt the NAOI as a covariate to alleviate the often-discussed problem of overdispersion in this type of models, the overdispersion remains an issue. One reason is the tendency of the model to underestimate the large daily precipitation amounts. This is because the model generates the precipitation amount using a transformed Gaussian process which reduces to a gamma distribution at individual locations. The gamma distribution is not a heavy-tailed distribution and is therefore not well suited to reproduce the heavy precipitation. However, Wilks
(2009) also found the underestimation of heavy precipitation in his study using a mixture of two exponential distributions to account for both smaller and larger precipitation amounts. This is another commonly reported problem for precipitation generators (e.g. Wilks (1999), Furrer and Katz (2008)). Allowing heavy-tailed distributions alleviates the overdispersion (Serinaldi and Kilsby (2014)), but simulating spatial extremes along with successfully capturing the smaller amounts with such a simple





model is even more challenging. Another way to reduce overdispersion is by allowing a suitable covariate such as seasonal total
precipitation in the GLM as shown by Kim et al. (2012) or by including seasonal dry/wet indicators as in Kim and Lee (2017).
Verdin et al. (2018) modified the model of Kleiber et al. (2012) by allowing the domain averaged seasonal total precipitation
as a covariate and showed that the inclusion of this covariate improved the simulation. However, their study was focused on
flat terrain where it is promising to take the areal precipitation as a covariate for the whole domain. This may not be suitable
for mountainous terrain and rather location specific climate information might be more promising. In this study, we wanted to
evaluate the model for its ability to reproduce the observed statistics at locations where no observations are available and for
that reason we avoided allowing any location specific information so that the model is not conditioned upon the availability of
the information at each grid point. We believe that, by allowing such gridded information as a covariate, the model performance
could be improved.

In complex mountainous terrain individual stations can exhibit precipitation characteristics quite distinct from those of neigh-
bouring (or more distant) stations with more typical characteristics. The station-by-station evaluation (Section 4.1) has revealed
that the model cannot reproduce precipitation at these distinct stations (e.g., Prutz in Figures 2–7). For the gridded simulations
we would have expected that these distinct stations might negatively influence their neighbouring stations. In contrary, how-
ever, the nearby stations have influenced the distinct stations by generating too strong correlations. Figure S4 (see Supplement)
depicts the inter-station correlations in the precipitation occurrence among the 29 stations in the observed against the simulated
data. The cloud of points having weak correlations in the observed data belong to the distinct stations, but in the generated data
the correlations are very strong. The reason is the good density of the network of 29 stations in such a small region. This is also
the reason that the largest discrepancy in the performance of both the occurrence and amount models amongst the 29 stations
is found at those distinct stations, because at those stations the model generates the statistics of the nearby stations rather than
reproducing their own observations. Thus, a distinct station in a region with dense observational network cannot be reproduced,
but doesn't strongly deteriorate the overall performance in the reproduction of the spatial information. As discussed in Section
3.2, this is due to the choice of the same set of covariates. Using the same set of covariates is, however, a necessary restric-
tion if the model is aimed at gridded output fields. Reproducing the spatial statistics over the whole region and especially in
mountainous region is indeed a challenging task. And with such a simple model, capturing the realistic spatio-temporal fields
of precipitation is even more challenging. Despite of that, our model successfully captures many difficult statistics useful for
climate change impact applications such as long spells of dry and wet days and areal monthly mean precipitation.

However, if a 'distinct station' is located in a data sparse area (such as St. Martin in our study area), it dominates the entire
neighbouring region and destroys the spatial structure. Thus, for a spatial precipitation generator in complex terrain the stations
should not only be selected according to data availability (and quality) but also based on their precipitation characteristics. If
they have distinctly different precipitation characteristics from the majority of the stations in the region, they should not be
included in the training data set and if one is explicitly interested in such a station, one should use a single-site approach.





Another limitation of this model is its inability to realistically simulate autumn and winter precipitation. This is because there
are systematic differences in the characteristics of weather types between various seasons. In autumn and winter, westerly cur-
rents are stronger and the associated precipitation patterns are more pronounced than during spring and summer. The precipi-
tation pattern in winter is associated with dynamically active synoptic-scale weather systems (fronts and low-pressure systems
specifically from the North Atlantic Ocean and the Adriatic Sea) in combination with orographic enhancement whereas the one
in summer is related to convective activity which is either embedded in frontal systems or generated locally. Our model does
not account for the wind-influence on the precipitation. This could be the reason for the model being not capable to capture the
spatio-temporal patterns in autumn and winter. The convective season in Austria usually starts in May and lasts till September
and during these months the model successfully captures the spatio-temporal patterns.

The covariance function used in the model is assumed to be stationary which may not be a realistic assumption. Detecting
spatial non-stationarity and modeling it is beyond the scope of this article and will be explored in future research. However, it
is possible that by considering a non-stationary covariance function (e.g. Paciorek (2003)), the model performance may be im-
proved. Since the precipitation-topography relationship is dominant in mountainous regions, the results can also be improved
by including not just elevation but other variables like slope, aspect, latitude and longitude in kriging (Wotling et al. (2000)).
Considering east-west and north-south gradients of the topography along with elevation in the KED also improves the results
(Hiebl and Frei (2017)). Since the inclusion of elevation at 1 km resolution has improved the results, we believe that consider-
ing even higher spatial resolution would provide even better simulation of precipitation.

## 6 Conclusion

A multi-site gridded precipitation generator that provides high resolution two-dimensional fields of precipitation in complex
terrain using historical observations from a network of meteorological stations is developed, implemented and evaluated. The
precipitation generator is an extension of the original framework of Kleiber et al. (2012) which uses a stationary isotropic
covariance structure. The original framework is based on a latent Gaussian process for the occurrence, and a transformed
Gaussian process for the amount of precipitation where gamma distributed random numbers are transformed to normally dis-
tributed random numbers. This framework considers the parameters of a Generalized Linear Model (GLM) as a realisation of a
spatial Gaussian process which allows one to spatially interpolate the parameters using kriging. In this article, two extensions
to the original framework are proposed: i) by allowing anisotropy in the covariance structure, and ii) by allowing elevation
as an external drift in kriging. The anisotropy is included in the model by taking into account the elevation difference in the
stationary covariance function of both, the occurrence and amount models. Along with that, elevation is allowed as an auxiliary
variable in the kriging equations for the interpolation of the parameters of both the occurrence and amount models.





The model is tested in a small region (about $100{\times}100$ km$^2$) with highly complex terrain in the European Alps where 29 observational stations with 30 years of data (1981–2010) are available. The test region comprises stations with elevation differences of about 2300 m. Thirty realizations of 30 years of synthetic gridded data at $1{\times}1$ km$^2$ resolution are generated to allow for a robust statistical assessment.


The main findings from this study can be summarized as follows:

- At individual stations where observations are available, the model reproduces the observed statistics realistically well, including annual cycles of daily probabilities of precipitation occurrence and monthly means of precipitation, dry and wet spells of different length and the overall distribution of precipitation amount.

- The model has a great capability to capture the spatio-temporal statistics in the complex terrain, which includes the spatial distribution of occurrence probabilities and amount, areal dry and wet spells of different length, monthly mean areal precipitation and monthly sum of areal daily mean precipitation.

- The proposed extensions considerably improves the simulation of spatio-temporal fields of precipitation – mainly due to the incorporation of elevation in kriging.

- The use of an isotropic or anisotropic covariance function in the mountainous region is equally good with marginal trade-offs for some of the statistics.

- The performance of the model varies greatly from month to month – being best in summer and worst in autumn.

- Intra-seasonal and inter-seasonal variabilities are well reproduced, while inter-annual variability is largely underestimated in autumn and winter.

- At a few of the 29 stations, where the observed precipitation statistics, and in particular their seasonality were distinctly different from all the other stations, the model performance is markedly compromised.

- The underestimation of large amounts of precipitation remains a problem.

Reproducing the spatio-temporal fields of precipitation in a region characterized by complex terrain like the Alps is a challenging task especially at locations where no observations are available. However, this is an essential requirement for hydrological 665 modeling as hydrological models are driven by spatially and temporally coherent precipitation data. The proposed model can respond to this need to some extent, nevertheless further improvement is required, as discussed in the article, to employ the model for downscaling purpose.





*Data availability.* The data used in this study can be available from the corresponding author upon request.

*Author contributions.* All the co-authors contributed to developing the idea. HD developed the code, prepared the input data, performed the simulations, did the analysis, prepared the visualizations and interpreted the results. MR supervised the work and MO provided vital suggestions. The manuscript was initially prepared by HD and finalized by all the co-authors.

*Competing interests.* The authors declare that they have no competing interest.

*Acknowledgements.* We are thankful to the data providers: the Austrian National Weather Service (ZAMG – Zentralanstalt für Meteorologie
und Geodynamik), Institute of Atmospheric and Cryospheric Sciences – University of Innsbruck, the Austrian Hydrographic Service, TIWAG (Tiroler Wasserkraft AG) and the Hydrographic Service of the Autonomous Province of South Tyrol.





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
