# Peer review of "A gridded multi-site precipitation generator for complex terrain: An evaluation in the Austrian Alps"

_Hydrology and Earth System Sciences, 2022_

## Author Comment (AC2)

**Authors response to comment on hess-2022-21**

**Anonymous Referee #1**

Author comment on **A gridded multi-site precipitation generator for complex terrain: An evaluation in the Austrian Alps** by Hetal Dabhi et al., Hydrol. Earth Syst. Sci. Discuss., https://doi.org/10.5194/hess-2022-21-RC2, 2022

This article suggests a novel GLM-based space-time rainfall generator for Alpine region. While the suggested model shows the limitations in reproduction of reality, I think this is a meaningful attempt in our field given the extreme challenging nature of the space-time field generation. The model is original, and the article is well structured. Therefore, I believe that the article is suitable for publication in this journal after a few revisions.

We are thankful to Reviewer-1 for the time and positive feedback on our manuscript. In the following we provide our response comment by comment.

1) I suggest authors to compare extreme values too. E.g. extreme values by the model of this study vs. AGPD, both point values and areal values. This is because one of the primary reasons of developing weather generators is to analyze disaster (from a probabilistic viewpoints).

   **Response:** We agree that one of the main purposes of developing weather generators is the analysis of extremes, but it is only one. This is the first time ever that such a model is used in complex topography to generate data at locations without historical observations. Therefore, we focused on testing our model in reproducing the basic statistical characteristics of precipitation over the complex topography. With respect to extremes, this at least includes the duration of wet and dry spells. We also have discussed the shortcoming of the model in reproducing some of the key statistics and concluded that further development is required in order to use it for a downscaling purpose. Therefore, including analysis of extremes will be a necessary next step, which is, however, out of scope for the present manuscript.

2) Please consider excluding trivial precipitation (e.g. less than 1mm) from your analysis to calculate occurrence-related values (Pww, Pdd, Pw, etc), and reanalyze your result. This is a common issue with all stochastic rainfall generator drawing rainfall depths from a modelled mathematical distribution. You may get better results.

   **Response:** We adopted the standard definition of 'wet day' used by the data providers and hence, when comparing to observations we have to stick to it. Thus, we have decided to leave the threshold as in the original manuscript.

3) L181-184. I would like to see the map of the interpolated scale and shape parameter. Interpolating parameter, in many cases, causes problems. The map should look smooth and should show dependency to the terrain. In addition, I suggest authors to consider obtaining these parameter maps based on the AGPD data or the KED-based rainfall map from your point observations to exclude the process of spatial interpolation.

**Response:** We have shown the shape and scale parameters in Figure R1 and R2, respectively (see below). The shape parameter is presented in Figure R1, which is constant throughout the year and hence only one plot, while Figure R2 displays the scale parameter for each month. As mentioned by the reviewer, the figures do show the dependency on the terrain. The dependency is more prominent in the warmer months for the scale parameter and less prominent and displays smooth fields during the colder months. This is reflected in the simulated precipitation amounts also. We will explicitly add this information to the manuscript in the revised version. Due to the satisfying quality of the interpolated parameters we have refrained from also testing any alternative as suggested by the reviewer.

[Figure]

Figure R1: Interpolated shape parameter

[Figure]

Figure R2: Interpolated scale parameter in each month

4) Figure 7 and Figure 18. I would also like to see the shades of the observed precipitations, which may be significantly greater than the current blue shades. This is not because I want to criticize, but because I would like you clearly show and mention the challenges of the stochastic rainfall generators (underestimation of large-scale variability) and to suggest potential remedies. Park et al. (2019) and Kim et al. (2020) discusses this issue in detail.

**Response:** This is indeed a relevant suggestion. However, the shaded region is the *tolerance interval* which cannot be determined for observed data. As opposed to confidence intervals, which would give expected bounds on the means of the simulated data, the tolerance interval gives bounds on the future individual observations. Here tolerance intervals are used to indicate the 99% range of the simulated values (with 95% confidence). In our view, tolerance intervals provide an appropriate visualization of the expected variability of the simulated data, as well as a means of comparison with the original data. We will consider adding this to the revised manuscript.

5) Figure 9. Why not show on the log-log axis? Too many small value pairs.

**Response:** In fact, we have started from the proposed representation as well. However, the advantages cannot compensate for the fact that the tolerance interval (the shaded region) on a log scale is hardly visible. We have therefore decided to stick to the linear scale.

6) Figure 12 and Figure 15 look like a collection of chessboards rather than a heat map. Would you remove the white squares?

**Response:** This is probably an issue of missed communication. We similarly noticed the 'chessboard-like' representation' of these figures *in certain browsers*. When we noticed this, we have added a comment regarding this on the HESS discussion page (https://doi.org/10.5194/hess-2022-21-AC1). We have corrected the issue with the figures now and it will be updated in the revised manuscript. We thank the reviewer for pointing this out.

7) Figure 11, 12, and 13 (Figure 14, 15, and 16 too): No need to show all the months. Please consider squeezing into one figure showing simulation ($1^{st}$ row), AGPD ($2^{nd}$ row), and differences ($3^{rd}$ row) for 4 seasonal months (columns).

**Response:** This is a nice suggestion. This would increase the readability of the article. We thank the reviewer for this comment. In the revised manuscript, we will make the necessary changes.

8) L 300-302, Figure 17. I am not sure which of the two variables that the authors are precisely comparing. Would you let me know how, for example, correlation coefficients were derived (e.g. x and y values of the scatter plot)?

**Response:** We considered the synthetic spatial-series of monthly mean of precipitation (MMP), i.e. the values of MMP at each grid point and similarly the values of MMP at each grid point in the observed data. Thus, the correlation coefficient is considered between the grid to grid values of MMP in the synthetic and observed data. Similarly, other error metrics have been derived for the corresponding statistics in the synthetic and observed data. A necessary change will be made in the revised manuscript making this clear.

---

## Author Comment (AC3)

**Authors response to comment on hess-2022-21**

**Anonymous Referee #2**

Author comment on **A gridded multi-site precipitation generator for complex terrain: An evaluation in the Austrian Alps** by Hetal Dabhi et al., Hydrol. Earth Syst. Sci. Discuss., https://doi.org/10.5194/hess-2022-21-RC2, 2022

The paper entitled 'A gridded multi-site precipitation generator for complex terrain: An evaluation in the Austrian Alps' by Hetal Dabhi and co-authors describes an extension of the daily stochastic spatio-temporal precipitation generator of Kleiber et al. (2012) to mountainous areas with complex topography, and illustrates the proposed framework in the Austrian Alps based on a network of 29 meteorological stations.

The paper is well written and structured, and the topic is relevant for the journal Hydrology and Earth System Sciences. The selected case study seems appropriate to test the proposed precipitation generator in an area with complex topography.

Despite the above qualities, the manuscript also contains major shortcomings, which, in my opinion, must be corrected before the paper can be considered for publication:

We are thankful to Reviewer-2 for the time and constructive feedback on our manuscript. In the following we provide our response comment by comment.

**Major comments:**

1) It is claimed that the paper provides an extension of the model of Kleiber et al. (2012) to areas with complex topography, but the proposed extensions (Kriging with external drift (KED) of model parameters and altitude dependent covariance function) are not very convincing. According to assessment results, these extensions do not offer significant improvements compared to the original model despite the addition of a lot of model complexity. In my opinion, the authors should therefore test the original model in their case study, and propose extensions only if the added complexity generates clear improvements in simulation results. If it is not the case, I would recommend to stick to the simplest possible model, and introduce the current paper as a case study testing the performance of the original model in the presence of complex topography, which I believe is already an interesting contribution.

**Response:** Concerning 'complexity of the model', it should be noted that we have proposed only two changes to the original model, which do not add much complexity to the model: i) Allow elevation dependence in the covariance structure, ii) KED instead of OK for interpolation of the parameters. Elevation dependence in the covariance structure is the natural assumption for the complex topography. Such elevation dependence has already been incorporated in the correlation structure by Wilks (1999)

and Wilks (2009). Due to the inclusion of the elevation difference in the covariance structure, only one parameter increases in each of the two models (i.e., the occurrence and amount models), which is the range parameter in the vertical direction. Also, we have explicitly stated in the results and conclusions that our model adds value due to the KED and not much due to the inclusion of the anisotropy. We also have implemented the original model (Iso-OK) in the same study domain but showed results only in Figure 19 and 20. We therefore do not see the need to make the original model the centre of our study. Actually, Figures 19 and 20 clearly suggest that the original model is indeed not a good choice especially for the amount model. We started with the hypothesis that anisotropy is required in the complex terrain and proved that it is not the case in the European Alps in such a small region. We have explicitly stated on L515 that Iso-KED is superior to the fully anisotropic case. This is also a choice of how the results are presented. One can start with the hypothesis that the original model is sufficient in the complex terrain and if not then propose the necessary changes. Or one can start with the hypothesis that the anisotropy is needed in the complex terrain which is a natural assumption and assess whether the hypothesis is correct or not. We have chosen the latter approach. However, we will mention the information on the complexity of the model in the revised manuscript.

2) Almost no information is provided about the spatial interpolation of model parameters (using different versions of Kriging, in particular KED), which is however a critical step of the model set-up. I think that it is inevitable to give more details about the Kriging step, including: (i) mention if a nugget term is used, and if yes what is the nugget contribution to the total variance, (ii) mention which variogram model is used, which method is used to fit or infer variograms, and maybe show some examples of adjusted variograms, (iii) prove by some data analysis than model parameters are (linearly) dependent of altitude to justify the use of KED, and finally (iv) display maps of kriged model parameters.

**Response:** We agree that the information provided on the spatial interpolation of the parameters may have been on the low side — this was due to the fact that there are quite a few parameters (and their number grows with the number of data points and is not fixed as in a parametric model). Based on the reviewer's comment we will add the following information to the revised manuscript.

i) will be mentioned in revised manuscript.

ii) An appropriate information will be added in the revised manuscript.

iii) Please see the plots below in Figure R1. The use of the KED is natural in the complex terrain. It has been used in the literature for precipitation interpolation in the mountains. We will add this information to the revised manuscript.

iv) Please see the plots for shape and scale parameters in Figure R2 and Figure R3. We will include this information in the revised manuscript.

[Figure]

Figure R1: The shape parameters (top) and scale parameters (bottom) of 29 sites plotted against their elevation

[Figure]

Figure R2: Interpolated shape parameter

[Figure]

Figure R3: Interpolated scale parameter for each month

3) A cross-validation is missing to evaluate the performance of the interpolation of model parameters by Kriging, and to assess the density of stations required for model calibration.

   **Response:** We thank the reviewer for the suggestion to add a cross-validation. We will include cross-validation in the revised manuscript. Following the 'logic' of our approach (i.e., to show exemplary results, such as in Figures. 2, 3, 4 where a 'good', a 'topographically exposed' and a 'bad' station are shown, we have decided to perform the cross-validation by holding those three stations out.

   However, the other suggestion by the reviewer, i.e. the assessment of the density of the stations required for model calibration is out of the scope of this study. We decided to include as many sites as possible and with any (high-quality) data, the more the better. Moreover, the observed data display that the spatio-temporal structure of precipitation is highly variable in mountainous regions, and in that regard, we believe that such an assessment would not be feasible.

4) The proposed model fails to reproduce precipitation at one of the three stations selected for illustration (i.e. Prutz), and according to Fig 9 also performs poorly for Dresdner Hütte, Kühtai, Nauders and Sankt Leonhard 2 (i.e. 17% of stations in total), and the reasons why the model fails at these locations are not

investigated in enough details. The only explanation given for Prutz is that this station exhibits 'precipitation characteristics quite distinct from those of neighbouring (or more distant) stations', but nothing is said about why a model calibrated at station locations is unable to reproduce precipitation at the exact same locations. For me it is very probable that the Kriging of model parameters introduces errors when clustered stations exhibit distinct statistics (which is the case for Prutz, that forms a cluster with Ried in Oberinntal and Fendels) - and possibly also where sharp gradients of precipitation occur (Nauders or St Leonhard im Pitztal 2) as well as at the edges of the study area (Marienberg, St Martin or Kühtai) - but one cannot see it in the present manuscript because the Kriging step is completely overlooked.

**Response:** We thank the reviewer for the quite detailed comment and substantiation with individual sites. We think, however, to have provided quite an extensive discussion on the problematic sites in Section 3.1 – and it is that in the complex terrain precipitation is highly variable in space and time and our data displays this behaviour. L588-596 explain the reasons for the bad performance at those sites. We have supported our argument by showing correlations in the observed and synthetic data at the 29 sites, which is shown in the Figure S4 in the Supplement. On L324–L328 (and also L588–L596) we explain the reasons, i.e. that the use of the same set of covariates at all the locations is the likely reason that the seasonality at those distinct sites is not well captured.

The 'probable reason' as pointed out by the reviewer has also been mentioned – see the explanation on L427–L431, which states that the kriging interpolation of the model parameters is the reason for the bad performance of the model [in certain parts of the domain]. We even explicitly discuss the case of the station St. Martin (L427–L429). We haven't discussed every station's performance individually in the manuscript, but collectively throughout the manuscript, we have mentioned the reasons for the bad performance at those sites.

5) Regarding the problem at Prutz station, I would be interested in seeing (in supplementary material for instance) the raw time series for Prutz, Ried in Oberinntal, Fendels and Ladis to be sure that the modeling problem does not simply originate from instrumental errors at Prutz station... the fact that this station has so much differences with Ried in Oberinntal which is located 1km apart, at the exact same altitude, and with similar neighboring topography is very surprising to me.

**Response:** Indeed, this was surprising for us too, but the data are from different data providers so the time of data recording, the method of data collections etc. may differ which results in data being different. We have carefully selected only those stations, which have gone through proper data quality check. Before selecting the stations for our study, we contacted each individual service provider and after getting an approval from them, we allowed only the high-quality stations in our study. Also, regarding Prutz and other stations showing very different characteristics than the surrounding stations, we again contacted the data providers to know the reason for the data being different as we also suspected that it could be due to instrumental errors or any other problem with the data. However, it was confirmed by the data providers that the data provided by them are of highest quality — and it should be not so surprising to see such behaviour in the mountainous region. To this end, we had the choice to either exclude those stations from the study or keep them as they are. We believe that 'throwing away' such stations from the study just for the reason that the data are not pleasant to the

model is not a good scientific practice. Hence, we allowed those stations — and decided to show them as 'problematic'.

**Minor comments:**

In addition to these major concerns, I also have some minor comments detailed hereafter. Line and figure numbers refer to: https://hess.copernicus.org/preprints/hess-2022-21/hess-2022-21.pdf.

1) that the value of 0.1mm corresponds to the lowest resolution of the rain gauges used in the case study. But I was not able to find this information. Please mention the resolution of the rain gauges in section 3.1.

   **Response:** We have adopted the threshold of 0.1 mm, which is the threshold typically used by the data providers and is also a common practice in the field. As per we know, a rain gauge does not have a 'resolution' as a thermometer.

2) L 125-128 and Eq4: Making the covariance of the occurrence altitude dependent seems a relatively arbitrary choice, and leads to a complex (and hardly tractable) model. This must be supported by some preliminary data analysis, showing that such altitude dependent covariance of precipitation occurrence actually exists in your study dataset. In addition, I wonder if including station altitude as a covariate in the vector Xo would not be a more convenient modeling choice. This may lead to a simpler model (also easier to 'validate' using the AIC/BIC model selection procedure introduced L260).

   **Response:** The altitude dependence of the parameters has already been raised in major comment (2). Please refer to the discussion there.

   Station altitude as a covariate is indeed an 'obvious choice' as a covariate — at first sight.  We originally also have considered including altitude as a covariate in the model, but at each location, this is just one number, i.e. a constant, while all other covariates are time-dependent, and the altitude doesn't change with time, so how does that one constant value influence the model? Or can it even be considered as a covariate if it is just a constant? Even if we allow altitude as a covariate, adding one covariate in the model would add one parameter at each location, so in total 29 parameters which after interpolation would increase to as many parameters as the number of grid points. In our view, the elevation dependence suggested by us makes the model simpler compared to considering elevation as a covariate.

3) Eq 5 (L156): It is not clear to me how this equation derives from Eq 2 and Eq 4 considered at a single site, and using previous day's occurrence as a covariate. Could you give more details (maybe in supplementary or a reference)?

   **Response:** Eq. 5 is not derived from Eq. 2 and Eq. 4, but it is stated that at individual location the model reduces to logistic regression. The idea is to use the logistic regression at individual stations with the selected covariates. One of the covariates is previous day's occurrence. This covariate is obtained from the observed data. Hence, when the logistics model is fitted at a station, the only location

dependent value is the regression parameter (β). Once the regression parameters are obtained at the 29 stations, a Gaussian process with the mean function as in Eq. 2. and the covariance structure as in Eq. 4 is implemented over the region. Such models, where previous day's occurrence is considered as a covariate already exist in literature. We already have given appropriate references on L148–L149 in the manuscript.

4) L166: Using KED with altitude as drift to interpolate regression parameters means that these parameters are all (linearly) correlated with altitude. This should be shown by a data analysis. In addition this leads to a complex model, and I wonder (as in my comment about Eq 5) if it would not be easier and equally effective to include altitude in Xo, and test if altitude is a relevant covariate.

   **Response:** Please see the answer for major comment (2) and minor comment (2).

5) Eq 6 and paragraph L179-188: the mean function of the latent process \mu_{A}, and the parameters of the Gamma distribution are space and time dependent, and in addition are interpolated by KED. This is a lot of parameters! You should quantify and acknowledge the complexity of your model. I'm not sure that so much model complexity (and degree of freedom) is necessary, but I'm ready to be convinced by a careful data analysis showing that all these dependencies are indeed present in your dataset. If i'm not mistaken, in Kleiber et al (2012), the mean function of the latent process Wa used to model precipitation amount is fixed to zero (and not regressed on covariates with regression parameters additionally interpolated by KED), which makes the model of precipitation amount way simpler. Such addition of complexity must be supported by data.

   **Response:** Indeed, the model has a large number of parameters because a Gaussian Process is a non-parametric method. As the number of data increases, the number of parameters also increases, which is different to the case of parametric models where the number of parameters remains fixed with increasing the data size (this is also the reason why the 'mean' and the 'covariance' are referred to as functions in Gaussian process modelling). We think that our model is equally as complex as that of Kleiber et al (2012). It has only two parameters more than that of Kleiber et. al (2012) — which are the two range parameters in the vertical direction due to the inclusion of the elevation dependence. Kleiber et al. (2012) presented the ***general framework*** for the multi-site gridded model but tested the model only for **multi-site** data generation and not for **multi-site gridded** data, i.e. it was tested at stations with observations and the interpolation part was not carried out and, in that sense, it was indeed less complex than our model. Also, please refer to our response to major comment (2) for the comment on dependencies.

6) Figure 1: I think station 22 (Pitztaler Gletscher) should be in red instead of station 24 (Obergurgl).

   **Response:** We thank the reviewer for pointing out this mistake. Earlier, we selected Obergurgl as a representative station for high mountain stations but later we changed to Pitztal Glacier and mistakenly updated the wrong figure.

7) L218-220: Could this extreme value be an outlier? Prutz is surrounded by very nearby stations (few kilometer apart) in all directions, and none of them measure more than 35mm this day (compared to

156mm in Prutz). I agree that summer convective rains can be very localized, but I'm still surprised by this observation, and I think this requires more investigation. And this also rises concerns about the quality of data at Prutz.

**Response:** Please refer to our response to major comment (5).

8) L223-224: different precipitation features at St Leohard 2: more details are needed to ensure that this station operates properly (same comment for Prutz).

**Response:** This station is operated by the Austrian Weather Service (ZAMG) and is properly quality controlled. Also, please see our response to major comment (5).

9) L229-232: I do not understand this paragraph. How this 7-days window increases the amount of data? And how simulation will add robustness to the observations? This is very unclear. Maybe because I do not understand what you name data.

**Response:** Since have only 30 years of observed data, there are only 30 values for each day (for example, January 1 has only 30 values). There are many days, for which all the 30 observations were dry, for example, so that the determined daily probability of dry days was 1. This would also then imply that every simulated realisation of those dates is a dry day, which is not realistic. There is always some — albeit possibly small — probability of precipitation on a particular day for the climate in the Alps. Similarly, for transition probabilities also some days (i.e., dates) were displaying the probabilities either 1 or 0. Hence, by allowing a 7-days window centred at the day of interest, we were able to escape the 1/0 values of the observed probabilities. We will explain this more precisely in the revised manuscript.

10) Eq 7 and Eq 8: A lot of temporal covariates are tested, but only one covariate linked to atmospheric circulation. Why such unbalance? I agree that NAOI can influence precipitation in the Austrian Alps, but it is definitely not the only covariate one can think about. In my opinion you should test other climate covariates, or justify why you think NAOI is enough.

**Response:** We agree that other covariates should have been incorporated in the model, however, at the time when we designed the experiment, we considered only NAOI as we wanted to see how NAOI alone is able to simulate the precipitation, especially during the months when NAOI is said to be linked with the precipitation. In a future study we will assess the impact of other covariates related to atmospheric circulation.

11) L288-291: More information about the APGD dataset is required to allow the reader to understand the main features of this reference dataset. In addition, it should be mentioned somewhere that APGD is not a perfect reference. Finally, the use of a 5km spatial resolution reference does not make it possible to assess the fine scale patterns generated by your 1km resolution model. Hence, all fine scale patterns seen in Fig 11 and Fig 14 may only be artifacts of using KED driven by altitude. This must be mentioned somewhere in the paper, or Fig 11 and Fig 14 should be aggregated at 5km resolution to avoid over-interpretation of the results.

**Response:** This is indeed an important suggestion. We will add more information about the APGD data in the revised manuscript. As per the figures showing 1 km resolution of synthetic data, we believe to show the generated data as they are. We respectfully disagree with the comment that Figure 11 and 14 should be aggregated to 5 km because this information would then be lost.

12) Fig 5 and Fig 6: Frequency -> Frequency (%).

**Response:** We thank the reviewer for pointing this out. However, it is not in %, but it should be mentioned that it is frequency 'per year'. It has been mentioned in Figure 20, but needs to be mentioned in other figures, too. We will make the necessary changes in the revised manuscript accordingly.

13) Fig 9: Very useful figure. It could be improved by: (1) using the same range of values for abscissa and ordinates, and for all stations. (2) Add station Id in addition to station names, and maybe order stations according to their Id (as in Table 1). (3) Mention in caption which quantiles are used (percentiles I presume).

**Response:** We thank the reviewer for this comment. This will indeed improve the readability of the figure. Necessary changes will be made in the revised manuscript.

14) Section 4.2: It would also be interesting to display q-q plots of areal daily precipitation amount.

**Response:** We included the q-q plot of the areal daily precipitation amount initially, however given the length of the article we had to exclude it. Also Figure 13 in the manuscript gives the flavour of how the plot looks like.

15) L420: The influence of topography on precipitation occurrence (and also amount) may be an artifact of the model. I don't say that it is the case, but just that you do not prove it in this paper.

**Response:** Indeed, when modelling environmental data, one should always consider the possibility that the result could be a model artifact. In case of precipitation in mountainous terrain, however, it is pretty well established that orography does not only receive precipitation — but it also triggers precipitation (cf. 'orographic precipitation' (e.g. Rotunno and Houze, (2007)) which, in a mountain range like the Alps, is responsible for the vast majority of precipitation amount). Even summer precipitation is triggered (convective initiation) by processes that are related to orography (slope, exposition). If a model then returns an orography influence, we think that this rather reflects a model quality than an artifact. Proving this, however is not overly simple.

16) Section 4.3: Results in this section prove that the anisotropic model is not relevant in the present case. They also show very few improvement when using KED instead of OK. When considering how much model complexity and arbitrary hypotheses about orographic precipitation enhancement are added with KED I wonder if a direct application of the model of Kleiber et al, (2012), with maybe altitude as a covariate (to be selected by BIC/AIC) would not be a better option.

**Response:** We agree that the anisotropic model has only a negligible/neutral impact on model performance (it does have a slightly positive impact on the frequency of dry spells, though), while the use of KED instead of OK has a small but consistent positive impact. We respectfully disagree, however, about the added complexity (which is nil) and the arbitrariness of the assumptions (see our responses to major comment (1) and minor comment (15) above). In our model comparison (Figure 19) we have also included the original model of Kleiber et al. (2012) — thus allowing to assess the impact. In this sense, we have demonstrated that using the original model would not be a better option (even if the difference is not overwhelming).

Furthermore, *even if the model had zero impact*, we think that demonstrating that an obvious extension for complex mountainous terrain, i.e. to include terrain information in one way or the other (i.e., the vertical dimension), does not improve model performance, would be a scientifically valuable contribution. Hence, other researchers could spare their time to 'try out' this obvious extension and focus on other possible improvements. If the impact is positive but not very large (as in the present case), we hope to pave the ground for further studies which can make this impact even larger.

17) L532-533: It would be interesting to compare the original and the extended model in a schematic, including the number of parameters involved.

   **Response:** Please refer to our response to major comment (1) and minor comment (5).

18) L678-827: many typos in the doi of the references.

   **Response:** We thank the reviewer for pointing out the typos. They will be corrected in the revised manuscript.

*References:*

1. *Simultaneous stochastic simulation of daily precipitation, temperature and solar radiation at multiple sites in complex terrain DS Wilks, Agricultural and Forest Meteorology 96 (1-3), 85-101.*

2. *Wilks, D. S.: A gridded multisite weather generator and synchronization to observed weather data, Water Resources Research, 45, https://doi.org/10.1029/2009WR007902, 2009.*

3. *Rotunno, R. and Houze, R. A.: Lessons on orographic precipitation from the Mesoscale Alpine Programme, Q. J. Roy. Meteor. Soc., 133, 811–830, https://doi.org/10.1002/qj.67, 2007.*

---

## Author Comment (AC4)

**Authors response to comment on hess-2022-21**

**Anonymous Referee #3**

Author comment on **A gridded multi-site precipitation generator for complex terrain: An evaluation in the Austrian Alps** by Hetal Dabhi et al., Hydrol. Earth Syst. Sci. Discuss., https://doi.org/10.5194/hess-2022-21-RC2, 2022

This paper discusses the results obtained using a space-time generator for daily rainfall in the Austrian Alps. This is an interesting topic and the paper is generally well written. The proposed model is very close to the one proposed by Kleiber et al. (2012) and the methodological contribution of the paper is rather limited. However, the authors have done an impressive work to validate the model on a complex data set. I think that this may be of interest for the readers of the journal Hydrology and Earth System Sciences. I suggest that the authors take into account the following comments before submitting a new version of their paper.

We are thankful to Reviewer-3 for the time and constructive feedback on our manuscript. In the following we provide our response comment by comment.

1) Line 3. "...as well as future climat ". Indeed, this would be great, but this is not discussed later in the paper. I think that it should be removed from the abstract or discussed in the paper.

**Response:** The future climate is mentioned in the opening of the article as to let the reader know the fact about the requirement of high-resolution data for current as well as future climate. In our view, the 'future climate' here simply refers to a potential application of a precipitation generator. Later in the abstract it is made very clear that 'this study aims at evaluating…' (which is not possible for future climate). In the revised version we will replace 'as well as' by 'or' to make this even more obvious.

2) Line 5 . "... propose an extension...". Please detail, contributions should be clear when reading the abstract.

**Response:** This will be considered and necessary changes will be made in the revised manuscript.

3) Line 38 "...typically of 1 km for spatial and daily for temporal scale". Could you precise an application where such scales are involved? I am not a specialist in hydrology, but I have the feeling that if you consider such a high spatial resolution, then the temporal resolution should also be finer?

**Response:** This is indeed a very valuable comment. We think, however, to have made exactly this assertion a couple of lines later (L41. '…in the spatial scale of 100 m and hourly for temporal scale.'). In the paper we propose an approach to generate 100 m (spatial) but daily (temporal) data — which might seem to be inconsistent, but is due to data availability (there are very few hourly data sets over

climate time scales available for training). Methods exist to 're-sample' daily variables to finer time scales (e.g. Delle Monache, et al. (2013)). We will add this information to the revised manuscript where the data is introduced.

4) Line 90. "Such WGs are of limited use if the observed gridded data are not available which is often the case". One option would be to "interpolate" the data on a grid before fitting the Wgs. Could you comment on the benefit of the proposed approach versus interpolation? It may be possible to obtain high quality gridded data using an interpolation method which merges all the available sources of information (meteorological stations, but also radar, models and so on), whereas the inclusion of such information in the proposed Wgs does not seem straightforward?

**Response:** Indeed, it is an option to first produce gridded (interpolated) data and then fit the gridded WG. This however, is a formidable task of its own, which is not just a little extension to our gridded precipitation generator. This essentially is another study (and for once we think 'being out of scope for the present study' is an appropriate wording). Therefore, if the gridded data does not yet exist, it is not an option. For Austria INCA data (the nowcasting facility of ZAMG, the Austrian Weather Service) at 1 km spatial resolution are available (which would be already good) — but only starts from 1999, unfortunately.

5) Line 100 "However, Kleiber et al. (2012) tested the model only for the multi-site precipitation generation, i.e. at locations with observation and not for the generated gridded data of precipitation." Does it make an important difference? If yes, please detail.

**Response:** Indeed, it makes a big difference. Since Kleiber et al. (2012) proposed the model that can generate gridded data of precipitation, but they tested their model only for the stations with observations, i.e. they tested only the **multi-site** model and not the **multi-site gridded** model. This means that the model was used to generate correlated data at stations with historical observations only and the interpolation step for producing gridded data was not carried out. Since many applications for impact studies require gridded data as input, before producing the gridded data for such applications, one must evaluate the model for its ability to reproduce gridded fields of precipitation.

Another important point is, since the model can provide data at locations without historical observations, one can obtain the historical time-series of daily precipitation at those locations. In order to use the model for this purpose, it is necessary to assess the model performance for the gridded fields.

We will add a sentence or two to clarify this in the revised manuscript.

General comment on Section 2. I find that the statistical methodology is not described precisely enough. The reader sometimes has to guess how the model is defined, and I do not think that there are enough details for someone who would be interested in reproducing the results. This is especially true in Section 2.2, but I think that more details should also be given in Section 2.1.

**Response:** We thank the reviewer for pointing this out. We will provide more details and make it more precise in the revised manuscript.

Please also

- explain how the model is fitted to the data, eventually provide the codes,

> **Response:** This will be added in the revised manuscript.

> - give the number of parameters involved in the model,

> **Response:** Gaussian process is considered a non-parametric method. In a parametric model, the number of parameters stay fixed with respect to the size of the data and it is easy to report the number of parameters. This is not the case with non-parametric methods where the number of parameters grows with the number of data points. We will include this in the revised manuscript.

> - comment on the computational time to fit and simulate the model.

> **Response:** This will be included in the revised manuscript.

6) Line 164 "The Gaussian process itself provides a spatial interpolation method 'kriging' so that the model parameters $\beta O$ associated with each covariate, which are estimated at observation locations, can be interpolated to any location of interest." Not clear for me, please reformulate.

**Response:** We thank the reviewer for pointing this out: we will reformulate this in the revised manuscript.

7) Line 230 "To reduce uncertainty and add more robustness to the observations, we increase the sample size of the observed data by considering a 7-days window centred at the day of interest." This sentence is mysterious for me, please reformulate.

**Response:** We will reformulate it in the revised manuscript.

8) Table 1. Is this table useful?

**Response:** We agree with the reviewer that Table 1 might not be useful if the stations were in regular terrain. In the present case, however, when the stations are in so different orographic settings, we think it might be helpful to keep the table as it is. In the revised manuscript we will therefore keep the table as it is in the original manuscript. Also, we have presented the results using the names of the stations, so it is easier for the reader to follow, if the details on the stations are given.

9) Line 267. "We select the covariates using both AIC and BIC …". If you consider AIC/BIC, then the model was fitted using Maximum Likelihood? Or only part of it?

**Response:** Yes, this is correct, a maximum likelihood approach was used for fitting the model. This will be mentioned in the revised manuscript.

10) General comment on Section 3.3. The authors have done an impressive work for validating their model. However, I have the feeling that the two following aspects are important but not discussed and thus should be further considered:

- Cross-validation. If I understand correctly, no cross-validation is performed although it is usually done when validating spatial Wgs. Cross-validation would consist in removing some stations when fitting the model and, then check if the model is able to generate realistic precipitations at these stations by comparing simulation and 'true' data. It may give confidence in the ability of the model to generate precipitation at locations where no data are available.

**Response:** We thank the reviewer for this comment. We will perform the hold-out cross validation and update the results in the revised version of the manuscript.

- Spatial dependance. The spatial dependence structure, which is an important aspect for many hydrological applications, is discussed only in the discussion (Line 585-610) and supplementary material. This should be discussed in Section 4.

**Response:** Indeed, spatial dependence of the hydrological aspects is only discussed in Section 5. This is due to an attempt not to mix the presentation of results (in a technical sense) and their discussion. We acknowledge that many readers might expect that discussion already in Section 4. In the revised manuscript, we will therefore add an opening sentence to Section 4.2 in which this expectation is directed to the 'Discussion' section.

11) Line 292. It is not clear for me why the authors use tolerance intervals instead of confidence intervals. Confidence (or fluctuation) intervals have the advantage of being widely used when validating Wgs and easily understood by most readers.

**Response:** Confidence intervals are sensitive to sample size. As the sample size increases the confidence interval narrows down, while tolerance intervals do not face such issue. This will be clarified in the revised manuscript.

12) Line 300. "To quantify the model performance…". Is it useful to have all these criteria? Do they bring complementary information? It takes space in the paper (with tables and plots) but it is barely discussed in the paper.

**Response:** Different evaluation metrics have different purposes. Showing one or two metrics only would not be sufficient to give the clear picture of the model performance to the reader. For example, RMSE is a commonly used error metric, however, showing only RMSE does not tell anything about the bias and for that reason we selected two other error metrics where MBE shows the positive or negative bias along with the absolute error MAE. Similarly, correlation shows the association between

the two variables but R-squared shows the variation. In the revised manuscript, we will add a brief description about performance metrics.

13) Line 395. The Kolmogorov-Smirnov test and the Wilkoxon-Mann-Whitney test are valid for continuous distributions, whereas rain gauge measurements are usually discrete (e.g. every 0.2 mm for tipping-buckets). Could you comment on the validity of the tests in such situation?

**Response:** The reviewer is indeed right that the two tests are valid for continuous data. We respectfully disagree, however, with the assessment that the (low) resolution of a rain gauge measurement makes it discrete (otherwise, every technical process imposing a resolution in any measurement would result in a discrete measurement). In fact, 'discrete data' are usually considered to be countable — but the fact that a tipping bucket 'counts' tipping points doesn't make this the principle of the measurement. The measurement principle is indeed given by a physical law (the weight of a droplet with some assumed density overcoming the known resistance of the 'tipping structure') which technically yields a relatively coarse resolution.

14) Line 496. "It is evident that by allowing the elevation as a covariate in the kriging interpolation for prediction at each grid point, the amount of precipitation is considerably improved". Is it so obvious? Maybe there are some improvements, but I don't have the feeling that they are 'considerable'!

**Response:** In our view, it can be called 'considerable' except for the months October to December. There is indeed a considerable improvement in the amount during summer months. We will take this into account in the revised manuscript.

15) General comment on Section 5 and 6. There are repetitions in Section 5 and 6. I suggest that you concatenate both Sections and try to make it shorter.

**Response:** Indeed, separating the two sections gives rise to a certain degree of repetition. It does, on the other hand, also clearly have advantages which we wouldn't want to miss. In the revised manuscript we will try to minimize the repetitions so as to shorten it at a little bit.

References:
1) Delle Monache L, Eckel FA, Rife DL, Nagarajan B, Searight K (2013). *Probabilistic Weather Prediction with an Analog Ensemble. Monthly Weather Review, 141(10), 3498{3516. doi:10.1175/MWR-D-12*-00281.1.

---

## Author Response (AR2)

**Authors response to comment on hess-2022-21**

We are thankful to Reviewer-2 for the time and feedback on our manuscript. In the following we provide our response comment by comment. We have made several changes according to the comments by the reviewer in the revised manuscript which are highlighted by magenta colour. The texts in blue colour are the changes already made in the previous round of the review process.

This is my second review of the paper entitled "A gridded multi-site precipitation generator for complex terrain: An evaluation in the Austrian Alps" by Hetal Dabhi and co-authors. My general opinion about the paper did not evolve much since my first review because, except for the implementation of a cross-validation experiment, most of my comments have been rejected without convincing counter-arguments.

To start with a positive note, I am glad to see the addition of a cross-validation experiment, which is convincing.

A more debatable aspect is the new statement (L 151) that "elevation dependence in the covariance structure is the natural assumption in complex terrain". I may accept "is a natural assumption" if more details were given (for instance by discussing in more details the references Wilks (1999, 2009)), but it is definitely not "the" natural assumption.

Beyond the choice of phrasing, I red with interest the two above references that are cited by the authors to justify the choice of an elevation dependent covariance. I acknowledge that in Wilks (1999) an altitude dependent covariance function is used, but I would like to draw your attention to the fact that in this study, the authors used a model selection step which leads to sometimes remove altitude from the covariance function of precipitation occurrence (2 months over 12), and most of the time for precipitation amounts (7 months over 12). In my opinion this shows that this kind of parametrization is fair (and I would like to emphasize here that I have no problem if you want to keep it), but should be handled with care (hence the necessity to discuss it thoroughly, and not take it as obvious). In addition, it should be noticed that in Wilks (2009), the same author does not use altitude in the covariance function of precipitation anymore, but keeps this parametrization for temperature only. Which makes sense to me, because vertical lapse rates are more obvious for temperature than for precipitation.

To sum up my opinion on this point: I am ready to be convinced that the proposed covariance function incorporating altitude is worth implementing in your context, but you need to put a bit more efforts in justifying why.

**Response:** First, we would like to mention that in Wilks (1999), the model is implemented in the complex topography with highest elevation of 2500m, so there is elevation dependence included in the covariance structure. Whereas in Wilks (2009), the model is implemented in a region with the highest

altitude of approximately 900m only and hence altitude was excluded from the covariance structure of precipitation. Elevation was included for modelling temperature in Wilks (2009) because, as the reviewer already have mentioned, temperature lapse rate has a sharp gradient.

Now, concerning the term 'the natural assumption', it is also a fact that elevation affects precipitation. It is well documented in literature that high altitude often receives more precipitation. One can find many such studies where precipitation modelling involved elevation in complex topography. Gafurov et al. (2006) showed the relationship between precipitation and elevation (also known as 'precipitation lapse rate'). Sasaki and Kurihara (2008) also showed precipitation lapse rate and pointed out that the correlation between precipitation and elevation is weak but statistically significant. Daly et al. (2008) pointed out that the relationship between elevation and precipitation is highly variable, but precipitation generally increases with elevation with exceptions when terrain rises above the height of a moist boundary layer or trade wind inversion. Thus, one can say that elevation is a significant predictor for precipitation in the mountains, especially in a complex terrain like the Austrian Alps where the highest elevation in our study area is 3533m a.s.l., it is natural to consider elevation as a predictor variable. Zhang et al. (2021) showed that the micro-physics also vary with altitude (due to a temperature and pressure dependence).

Also, we already have given two references on line 176, where the authors used elevation for precipitation interpolation in the mountains and also showed that KED outperforms OK. In Hiebl and Frei (2018) one can find the use of KED for constructing high-resolution (1 km) gridded daily precipitation climatology for Austria which indeed includes our study area. Hence, including elevation for precipitation modelling in the mountainous region is well 'a natural assumption'. We admit that the use of the article was wrong in the sentence and it should be 'a natural assumption' and not '*the* natural assumption'. This is corrected in the revised manuscript on line 152.

We hope our answer justifies the use of elevation in the covariance structure.

**Finally, there are several aspects that I still find problematic:**

**1)      I have the impression that the absence of nugget in the Kriging step may generate artifacts, and your response to my comment as well as the associated changes in the manuscript (which are restricted to acknowledge the absence of nugget) are not convincing.**

**Kleiber et al. (2012), which was the starting point for the present study, used a nugget effect when kriging model parameters. You decided to remove this nugget. This may be Ok (I honestly don't know), but you have to justify it. This can be done either by explaining why do you think there is no small scale variability in the parameters to interpolate, or (even better) by showing that these parameters do not display small scale variability. For the later option, the best solution is probably to show that the empirical variograms of the parameters interpolated by Kriging are close to zero for short lags.**

**Response:** We would like to point out that the nugget has not been forced to be zero but rather it is estimated as a very small value – nearly zero which we have allowed as it is. Since the reviewer suggested to give information on the nugget term, we mentioned it. However, with the changes in the manuscript where we have added information on the variogram. It has been estimated using MLE and the estimated value albeit close to zero is what we have allowed in the model. We do not think it is necessary to show the variogram in the article. However, for an example we present one of the variograms here (Figure R1).

Figure R1: Variogram cloud for one of the regression parameters in the GLM for occurrence

Necessary changes have been added in the revised manuscript and have been highlighted on L168-170, L180-184 and L186.

2)      The figure R1 displaying scale paramater=f(elevation) and shape parameter=f(elevation) in the response to my major comment 2 (in the response to reviewers file) does not show any clear relation between the parameters to interpolate and the external drift, and therefore does not really justify KED.

In the same line, I don't think that the new figures 16 and 17 (formerly Fig 19 & 20) show a clear advantage of KED vs OK, nor of isotropic vs anisotropic covariances. This is in line with the comment 14) of Reviewer#3: "Maybe there are some improvements, but I don't have the feeling that they are 'considerable'!", and the changes made in the manuscript (L 567) are in my opinion too        minor        to        reflect        this        shared        concern. I therefore strongly recommend to the authors to rewrite the discussion of Fig 16 and 17 to clearly acknowledge that the improvements associated with the use of KED and anisotropic covariance are rather limited.

**Response:** We agree with the reviewer that the relationship between the parameters and elevation is not strong. As described in our aforementioned response to the main comment, Sasaki and Kurihara (2008) pointed out that the correlation between precipitation and elevation is weak but statistically significant. Please read our response to the main comment. Regarding the use of the anisotropic covariance, we already have acknowledged the performance of the model in the conclusion on L732-L733 in the revised manuscript.

For Figure 16, we agree that saying KED outperforms OK in ALL the months can be criticized. Hence, we have modified the sentence and highlighted the change in the revised manuscript on L570-L573.

For Figure17, it is noteworthy that the presented statistics for dry/wet spells is a frequency **per year**. Since the anisotropic model has a slightly positive impact on the frequency of dry spells, an improvement of say, 2 dry spells per year means in 30 years of data there will be a difference of 60 spells which is indeed not negligible. Therefore, in our view, the model performance can be said *better*.

**3)      The explanation about the absence of outliers at Prutz station (in the response to reviewers file) and the reason why you don't want to investigate it further are not convincing (that      is      the      least      I      can      say).  Why don't you want to display the data to understand what is going on? Either the data from this station is an outlier, and removing it will improve the performance of your model. Or there is a micro-climate at this location, and this is interesting to point out. In any case this must be discussed in more details.**

**Response:** The reviewer stated in their question in the first review that "*for Prutz, Ried in Oberinntal, Fendels and Ladis to be sure that the modeling problem does not simply originate from instrumental errors at Prutz station...*"

In our answer, we have emphasized the fact that all the stations in our study are of high quality, so we (again) stand by our response that there is no scope for claiming that the data could be erroneous.

Now, as for the precipitation characteristics at Prutz being dramatically different from the surrounding stations, it is really interesting to investigate the reason for that, but that is not the goal of our study and is indeed out of the scope of the study. We have spent substantial amount of time over this station to figure out the reason for it being an outlier. We also would like to point out that the reviewer writes "***Either the data from this station is an outlier, ...",*** this statement is not clear to us (whether the whole data set meant or just one datum). However, we assume that the reviewer meant the whole dataset from Prutz is an outlier.

We would like to point out that we already have removed this station in the cross-validation experiment and showed that the results are not affected by this station. This is also one of the findings of our study that one 'outlier station' is not affecting the results if it is surrounded by a good density of observation locations. Contrary to that, an outlier station can affect the spatial structure in the generated data if it

is in a sparse network of observations which is the case with the station St. Martin in South Tyrol. In our view, these two are important findings which we have stated in the manuscript on L663 to L673 and which can be useful for other researchers in their analysis. Also, we again emphasize that removing a station away from the study because it exhibits different characteristics from other observations is not a good scientific practice — especially in the mountainous region where precipitation has a highly variable nature.

As for the comment about displaying the data for investigating the reason for Prutz being an outlier, we don't see the need for displaying the raw data because looking at the raw data one cannot investigate the reason for the presence of the outliers or micro-climate in the data. We also do not neglect the possibility of micro-climate at the location. It is not that we didn't make any attempt to investigate the possible reasons for having outliers. If the data had revealed any possible reason/s for the presence of outliers during our investigation, then it would be one of the important findings and we certainly would have discussed it with graphical illustration in the manuscript. However, we are displaying the raw data here for Prutz and its one of the nearest sites Fendels in Figure R2.

[Figure]

[Figure]

Figure R2: Raw timeseries (1981-2010) of precipitation observations [mm] at Prutz (upper) and Fendels (lower)

**4)** Your response to my minor comment 10) (**i.e. why using only NAOI as covariate linked to atmospheric circulation) is not satisfactory.**

**Response:** As pointed out in our original reply, it was a *choice* to consider NAOI as an exemplary variable for this study. Indeed, there are other good options (as suggested by the reviewer in their original review) – and this will be investigated in more detail in the future.

**5) Your response to my minor comment 2) is not satisfactory. Most of your covariates for Xo(s,t) are functions of t only (hence constant through space) (e.g., cos(2\*pi\*t/n), sin(2\*pi\*t/n), etc.). What I propose is to add a covariate that is function of s only (hence constant through time). Xo(s,t) is indexed in both space and time. I hope that put this way you can see the symmetry of the problem in space and time, and then the possibility of using altitude as a covariate.**

**Response:** We thank the reviewer for elaborating their point. We see the reviewer's point. Considering altitude in the GLM will allow the model to include spatial information but only in vertical direction. However, that would not be sufficient for considering the covariates as a function of space. In our view, along with elevation, there could also be latitude and longitude and if possible other topographic information, e.g. slope or exposition, allowed in GLM.

However, based on our experience and preliminary analysis, we can say that it (i.e. inclusion of elevation) would not add considerable improvement in the results – at least not 'more considerable' than KED. We will consider this suggestion in our future work.

*References:*

1) *Daly, Chris & Halbleib, M. & Smith, Joseph & Gibson, Wayne & Doggett, Matt & Taylor, George & Curtis, Jan & Pasteris, Phillip. (2008). Physiographically-Sensitive Mapping of Temperature and Precipitation Across the Conterminous United States. International Journal of Climatology. 28. 10.1002/joc.1688.*

2) *Hiebl J., Frei C. (2018): Daily precipitation grids for Austria since 1961−development and evaluation of a spatial dataset for hydro-climatic monitoring and modelling. Theoretical and Applied Climatology 132, 327-345, doi:10.1007/s00704-017-2093-x*

3) *A. Gafurov, J. Götzinger, and A. Bárdossy, Hydrol. Earth Syst. Sci. Discuss., 3, 2209–2242, 2006 www.hydrol-earth-syst-sci-discuss.net/3/2209/2006.*

4) *Zhang, Z., Song, Q., Mechem, D. B., Larson, V. E., Wang, J., Liu, Y., Witte, M. K., Dong, X., and Wu, P.: Vertical dependence of horizontal variation of cloud microphysics: observations from the ACE-ENA field campaign and implications for warm-rain simulation in climate models, Atmos. Chem. Phys., 21, 3103–3121, https://doi.org/10.5194/acp-21-3103-2021, 2021.*